# ON UNDERSTANDING OF THE DYNAMICS OF MODEL CAPACITY IN CONTINUAL LEARNING

## ABSTRACT

The core issue in continual learning (CL) is balancing catastrophic forgetting of prior knowledge with generalization to new tasks, otherwise, known as the stability-plasticity dilemma. We argue that the dilemma is akin to the capacity (the networks' ability to represent tasks) of the neural network (NN) in the CL setting. Within this context, this work introduces "CL's effective model capacity (CLEMC)" to understand the dynamical behavior of stability-plasticity balance point in the CL setting. We define CLEMC as a function of the NN, the task data, and the optimization procedure. Leveraging CLEMC, we demonstrate that the capacity is non-stationary and regardless of the NN architecture and optimization method, the network's ability to represent new tasks diminishes if the incoming tasks' data distributions differ from previous ones. We formulate these results using dynamical systems' theory and conduct extensive experiments to complement the findings. Our analysis extends from a small feed-forward (FNN) and convolutional networks (CNN) to medium sized graph neural networks (GNN) to transformer-based large language models (LLM) with millions of parameters.

## 1 INTRODUCTION

Humans can easily adapt to multiple tasks. However, when neural networks (NN) seek to mimic this behavior [49], they exhibit a phenomenon known as catastrophic forgetting, where the model forgets older tasks while learning new ones [49]. This well recorded issue is seen irrespective of the NN architecture, from simple linear adaptive systems [33] to massive large language models [46, 37]. The field of artificial intelligence that studies this phenomenon is known as continual learning (CL).

In recent years, numerous studies in CL [14, 50, 30, 6, 45] have shown that the core issue behind CL is a trade-off between forgetting prior information (catastrophic forgetting) and learning new information (generalization), also known as the stability-plasticity dilemma. This trade-off captures the relationship between data and the optimization procedure, but that is only part of the picture. Independent lines of inquiry have also shown that over-parameterized NNs play a crucial role in achieving optimal performance in the CL paradigm [39, 21, 20]. While, [40] study the role of optimization characteristics, [27] study the learnability of CL problem when subsequent distributions are overlapping. Although, all of these works provide different but overlapping insights, they look at the different sides of the problem such as model and data in [31] or the model and optimization procedure in [27, 40, 6] and do not consider the complex interplay between the model/optimization/tasks.

In this work, we aim to provide holistic insights into this interplay and establish a foundational understanding of the effect of NN capacity (stability plasticity balance point) on CL optimization in the presence of a series of tasks. We extend the definition of capacity from [41] to the CL paradigm, describing capacity (Def 3) as the effect of network architecture, hyperparameters, and weights measured through the cost function. We then elucidate the connection between capacity and the stability-plasticity balance point (Lemma 1 and Fig. 1) and show that this balance point is akin to the networks ability to represent tasks - higher the capacity (measured through forgetting cost) lower the representation ability of the network. With this theoretical framework, we show that the smallest possible change in the network's capacity is a function of changes in weights and tasks (Theorem 1). Our main results (Theorems 2, 3) demonstrate that capacity, and by extension, the balance point, is non-stationary in the CL setting. The key conclusion of this work is:

*"The CL capacity of a model is a function of the interplay between model, data, and the optimization procedure. Moreover, regardless of the type of NN, optimization procedure, or task, the network eventually becomes unsuitable for representing the tasks if each subsequent task differs from the previous one even by a small constant."*

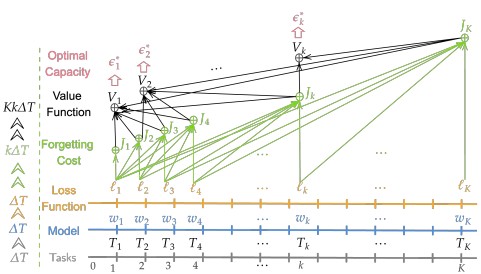

Figure 1: Visualizing the dynamic-program based CL formulation: For task $k$, the forgetting cost, $J_k$, is the expected loss over tasks $[1, k]$ (green arrows). The value function, $V_k$, is the forgetting cost over all tasks $[1, K]$ (black arrows). The optimal capacity, $\epsilon_k^*$, is the sum of effective capacities over tasks $[1, K]$ (in red).

The goal of this work is to elucidate the dynamical behavior of the representation power of the NN in the CL setting as a function of the network, optimization and the data. Thus, we are not proposing a new method or show performance improvement. Specifically, we show that, for different architectures, a small change in the tasks propagates through the learning problem to affect large variation in the model capacity (see Fig. 1, left side). We therefore, choose well-used datasets and vary the network architecture. In particular, we validate our theoretical results through four case studies. In the first case, using a synthetically generated sine wave dataset [25], we show that the capacity of a feed-forward NN (FNN) diverges even when the two major classes of CL methods, *experience replay and regularization approaches* are utilized. In the second, we extend the study to a standard convolutional NN (CNN) with the Omniglot dataset [2]. We also show results with a graph neural network in third case study and finally, develop a detailed study using large language models (LLM) to demonstrate our results. Our findings confirm that our theoretical results hold even when we scale from a simple FNN to a 134 million parameter LLM. Proofs complete with all assumptions are provided in the supplementary files.

## 2 RELATED WORKS

Starting from [19] in 1999 to [20] in 2024, numerous works have attempted to model/reduce catastrophic forgetting in neural networks. A simple taxonomy of recent published works reveals four categories: regularization-based [4, 28, 43, 30], model architecture-based [1, 10, 12, 16, 22, 31, 51], experience replay-based [7, 23, 29, 38, 52] and other optimization approaches for CL efficiency [11, 48, 53, 55]. This huge body of work is focused on improving empirical performance.

On the other hand, empirical attempts to study the characteristics of the CL problem have been made as well [14, 24, 32, 37]. For instance, [14] study the loss of plasticity in CL whereas [24, 32] study a phenomenon known as stability gap frequently observed in CL methods. The empirical investigative studies cover a wide range of neural network architectures as well, going from FNN/CNN in [14, 24, 32] to large language models in [37, 46]. Despite such a huge body of literature, there have only been a few attempts to study CL from a theoretical standpoint. The key reason behind this is that the NN learning problem in CL domain is rather complex to study requiring stringent assumptions that are scarcely held in practice. This is clearly seen from the few approaches that do theoretically analyze the problem. For instance, works in [20, 21, 15] study the effect of over parameterization and task similarity on forgetting with a linear model under two tasks. Catastrophic forgetting in the presence of task similarity in analyzed in the NTK regime in [13]. On the other hand [34] and [35] study the complete CL problem with a linear two layer NN. To the best of our knowledge, the only approach that does not make either a two task assumption or assume linearity of the model is [27] but instead focuses on the class incremental setting.

To obviate the necessity for such assumptions and provide a general framework to analyze CL, we take a Lyapunov analysis standpoint, a tool that has been used in the control literature [5]. In contrast with the existing literature, we analyze the CL problem through a dynamic programming-driven optimal control point of view following the perspective from [45]. The only assumptions required are twice differentiability and Lipschitz continuity of the loss function- two very practical assumptions in the NN learning domain and our analysis extends to a series of tasks. In a similar vein to [39] we also perform Taylor series approximation to get this differential equation characterization, however,

our theoretical analysis easily extends from a simple FNN to a llm- a very novel contribution to the CL literature. To the best of our knowledge there has been no theoretical study, where the analysis considers a dynamical behavior of the CL problem that extends across FNN/CNN/GNN and LLM.

## 3    Continual Learning Effective Model Capacity (CLEMC)

Let $x$ and $y$ be random variables corresponding to input and output probability spaces with support $\mathcal{X}$ and $\mathcal{Y}$ and $\mathcal{B}(\mathcal{X})$ and $\mathcal{B}(\mathcal{Y})$ representing the corresponding Borel algebras. Define $t$ as a random variable denoting the joint space of $x \times y$ with a model $f_{(w,h)} : \mathcal{X} \to \mathcal{Y}$ being specified using weights $w$ and hyperparameters $h$. Given compact sets $\mathcal{W}$ over $w$ and $\mathcal{H}$ over $h$, the goal is to learn the weights by searching over the hypothesis space $f = \{f_{(w,h)}, \forall h \in \mathcal{H}, w \in \mathcal{W}\}$ through a loss function $\ell_{w,h}(t)$. In this paper, we will assume that the hyperparameter/architecture is fixed and therefore, will drop the notation $h$ and denote loss simply as $\ell_w(t)$. Throughout the paper, we will assume $x_k = x(k)$ and use them interchangeably, and $\mathbf{k} = [1, 2, \cdots, k]$. In this context, we characterize the effective model capacity as follows.

**Effective Model Capacity:** We will assume that $\ell_w(t)$ is continuous and twice differentiable over the support $\mathcal{X} \times \mathcal{Y}$ or $\mathcal{X}$, and the compact set $\mathcal{W}$. Under these assumptions, let $\ell_{min} = \mathcal{O}_{\mathcal{W}}(T) = min_{w \in \mathcal{W}} \quad E_{t \in T}[\ell_w(t)]$ be the optimization procedure with $T$ being a dataset of samples $t$ with $T \subset \mathcal{B}(\mathcal{T})$. Then, given the best hyperparameter/architecture configurations, the optimization procedure $\mathcal{O}_{\mathcal{W}}$ seeks to find the weights $w^* \in \mathcal{W}$ that minimizes the loss over a dataset. Given this setting, we define the effective model capacity (the upper/lower bounds derived in the appendix) as the smallest achievable loss value using $\mathcal{O}_{\mathcal{W}}$ that remains unchanged even when additional data or training is used.

**Definition 1** (Effective Model Capacity (EMC))**.** *Given $\mathcal{W}$ as the weight space and $T \in \mathcal{B}(\mathcal{T})$ with an optimization procedure $\mathcal{O}_{\mathcal{W}}(T)$, the EMC of the model $f$ is given as*

$$\epsilon = \min_{T \in \mathcal{B}(\mathcal{T})} \left[\mathcal{O}_{\mathcal{W}}(T)\right] = \min_{T \in \mathcal{B}(\mathcal{T})} \left[\min_{w \in \mathcal{W}} \; E_{t \in T}[\ell_w(t)]\right] \qquad \text{(EMC)}$$

Def EMC takes an approximation error perspective (as in [42]), however, unlike [42], (EMC) depends on the optimization procedure, the model performance and the dataset. It is also similar to the capacity definition in [41], with the key distinction being that [41] focuses on the number of data points that are properly represented by the model. However, this way of defining capacity is often inadequate because numerical superiority over samples alone (without considering the data distribution characteristics) doesn't ensure model usefulness [17]. Since, a CL problem requires careful attention to the distribution characteristics, we define capacity through the forgetting loss.

**Characterizing the CL Balance Point:** CL involves learning a sequence of tasks indexed by $k \in [1, K], K \in \mathbb{N}$, where a task $k$ is represented by its dataset $T(k)$. The collection of all tasks until $k$ can then be denoted as $\mathbf{T}_k = \{T(1), T(2), \ldots, T(k)\}$ with $\mathcal{T}_k$ being the cumulative support. Given a feasible weight set $\mathcal{W}_k$, and loss function $\ell_{w_k}(t), t \in \mathcal{T}_k$, the model at $k$ is denoted by $f_{w_k}$, the goal of CL is to maintain memory of all observed tasks, then, the CL forgetting cost for the interval $\mathbf{k} = [1, k]$ is given as

$$\min_{w_k \in \mathcal{W}_k} J_{w_k}(\mathbf{T}_k) = \min_{w_k \in \mathcal{W}_k} \sum_{i=1}^{k} \gamma_i \left[\underset{t \in T(i)}{E}[\ell_{w_k}(t)]\right] , \quad \forall T(i) \in \mathbf{T}_k, \qquad (J_F)$$

where, $\gamma$ ensures boundedness of $J_{w_k}(\mathbf{T}_k)$ (see [45], Lemma 1). The growth of forgetting cost over progressively increasing task intervals (as new tasks arrive) is shown in Fig. 1 (in green). The forgetting cost formulation in $(J_F)$ is the standard in the CL literature [40] but, has two key limitations [6, 21] that we highlight using the following illustrative example.

**Example 1.** *Consider three learning tasks with feasible regions $\mathcal{W}_1, \mathcal{W}_2,$ and $\mathcal{W}_3$, centered at ideal solutions $w_1^*, w_2^*,$ and $w_3^*$. The naive cost setup in $(J_F)$ ignores the following interactions.*

***Sequential Optimization:*** *Solving the first task (attaining $w_1^*$) means the second task must start from $w_1^*$. Therefore, $w_1^*$ and its distance from $\mathcal{W}_1 \cap \mathcal{W}_2$ (the feasible region all solutions that work on both tasks 1 and 2) determines how close we can get to $w_2^*$. In general, as the optimal solution for tasks $[1, k-1]$ is used as the starting point for task $k$. the feasible region of the previous tasks has an influence on the subsequent task [15][Theorem 3.1].*

***Influence of future tasks:*** *If the second task induces a significant deviation from $w_1^*$, large forgetting is seen (see [15], Figure 1). Conversely, if the new task has no influence, there's no generalization.*

It is clear with this example that each tasks' solution has an influence on the future task and at the same time, future tasks performance dictates how well the the model can do on the present tasks. That is, there is an interplay between future tasks and the present task. Mathematically, a complete CL [45] characterization must therefore consider both the sequential optimization over tasks as well as how each tasks' solution impacts future tasks. For a fixed $h \in \mathcal{H}$, the complete CL problem is

$$V^{(*)}(u_k) = \min_{u_k} \quad \sum_{i=k}^{K} [J_{w_i}(\mathbf{T}_i)], u_k = \{w_i, i = k, k+1, \cdots K\} \quad \text{(CL)}$$

The optimization problem in (CL) provides the value function, where previous tasks are perfectly remembered (optimizing the sum of forgetting loss, $(J_F)$) and future tasks will be perfectly learnt (for task $k$, optimizing also for $[k+1, \ldots, K]$ via successive update of model weights). That is, given a starting weight set $w_1^* \in \mathcal{W}_1$, the solution to the CL problem with $K$ expected tasks is $\{w_1^* \in \mathcal{W}_1, w_2^* \in \mathcal{W}_1 \cap \mathcal{W}_2, w_3^* \in \mathcal{W}_1 \cap \mathcal{W}_2 \cap \mathcal{W}_3 \cdots w_k^* \in \cap_{k=1}^{K} \mathcal{W}_1\}$ and $V^{(*)}(\{w_1^*, w_2^*, w_3^*\})$ is the total cost (corresponding to the balance point). Naturally, the value of $\ell_{min}$(see Def 1) corresponding to each of these $w_i^*, i = 1, 2, 3, \cdots, K$ describes how well the model performs at the respective stages of the CL problem and therefore (summation of the losses) quantifies capacity in the CL setting. The value function and its progressive evolution is also illustrated in Fig. 1 (in black). At each task $k$, the value function considers all the previous tasks (arrows adding forgetting costs from prior intervals) and all future tasks (arrows adding forgetting costs from future intervals). We now extend Def 1 to define effective model capacity for a CL problem.

**CL Effective Model Capacity and Balance Point:** For ease of exposition, we begin by stating

**Definition 2** (Forgetting Effective Model Capacity (FEMC)). *For task $k \in [1, K]$, dataset $\mathbf{T}_k$, weight space $\mathcal{W}_k$, optimization procedure $\mathcal{O}_{\mathcal{W}_k}(\mathbf{T}_k)$, EMC at $k$, $\epsilon_k = min_{\mathbf{T}_k, w_k} J_{w_k}(\mathbf{T}_k)$, we define FEMC at task $k$ as:*

$$FEMC(k) = \max_{\mathbf{k}} \epsilon_{\mathbf{k}} = \max\{\epsilon_1, \epsilon_2, \cdots, \epsilon_k\} \quad \text{(FEMC)}$$

$FEMC(k)$ at each $k$ is defined by the highest forgetting loss in the interval $[1, k]$. For example, in a three-task scenario, the FEMC at task 3, $FEMC(3) = \max\{\epsilon_1, \epsilon_2, \epsilon_3\}$, and is determined by the task the model forgets the most. We now define CL effective model capacity as follows.

**Definition 3** (Effective Model Capacity for CL (CLEMC)). *For a task $k \in [1, K]$, we define CLEMC as the sum of FEMC across all possible tasks as*

$$\epsilon_k^{(*)} = \sum_{i=k}^{K} FEMC(i) = \sum_{i=k}^{K} \max_{\mathbf{i}} \epsilon_{\mathbf{i}} \quad \text{(CLEMC)}$$

Def (3) is closely related to the forgetting loss through FEMC. If the model learns multiple tasks, we initially obtain the FEMC corresponding to each task, and then, the $\epsilon_k^{(*)}$ is the sum of individual task FEMC (illustrated in Fig. 1 (in red)). Since the individual task FEMC is proportional to the loss function, perfect representation of the underlying tasks is implied by $\epsilon_k^{(*)} = 0$ and representation (and FEMC) gets poorer and poorer as $\epsilon_k^{(*)}$ increases. Notably, $\epsilon_k^{(*)}$ measures the models' CL performance.

Similar to (CLEMC), the measure of models' performance has also been defined proportional to the value of the forgetting loss. For instance, [27][Def 3.1] defines learnability as the gap between empirical risk and the smallest risk in the hypothesis space, but without the minimization over different data samples. Furthermore, [26][Theorem 1] suggests that necessary and sufficient conditions for good CL are proportional to effective learning on prior tasks, defined through the forgetting loss. In contrast with the above, where just loss on the prior tasks is considered, in Def (3), both future tasks and bias due to subsequent solution are also considered. The relationship between (CL) and Def. CLEMC is formalized in the next lemma.

**Lemma 1.** *For $k \in [1, K]$, let $u_k = \{w_i, i = k, k+1, \cdots K\}$ be weight sequences from $k$ with $\mathcal{U}(k) = \{\mathcal{W}_i, i = k, k+1, \cdots\}$– the compact sets. Next define $(J_F)$, (CL) and (CLEMC) to write*

$$\epsilon_{k+1}^{(*)} - \epsilon_k^{(*)} = min_{\mathbf{k}} \{\max_{\mathbf{T}_i} \{\langle \partial_{w_k} V^{(*)}(u_k), dw_k \rangle + \sum_{T \in \mathbf{T}_k} \langle \partial_T V^{(*)}(u_k), dT \rangle\}\} \quad \text{(FD)}$$

*Proof.* See Appendix B $\qquad\qquad\square$

*If each subsequent task is different than the previous task, the cumulative change in tasks, $dT(k)$, is going to lead to deteriorating capacity. In particular, the change in $dT(k)$, is going to drive a change in weights, $dw_k$, which in turn drives a change in capacity. This interplay is going to accumulate as the number of tasks increases and lead to deteriorating capacity.*

## 4 ANALYSIS

In this section, we perform a two-fold analysis to prove our main idea, "capacity diverges if tasks change constantly". First, we formally prove this result. Later, we demonstrate experimentally, that the capacity diverges irrespective of the model architecture or the data used. *An experimentally inclined reader can safely skip the theoretical analysis and get the same insights from our empirical observations.* We recommend reading this section to get an understanding into why capacity divergence occurs.

### 4.1 THEORETICAL ANALYSIS

We begin by deriving a lower bound on the first difference of $\epsilon_k^{(*)}$ (derived in Lemma 1)and then analyze the impact of the independent terms of the bound on the effective capacity.

**Theorem 1.** *The first difference in CLEMC* (FD) *is lower bounded as*

$$\epsilon_k^{(*)} - \epsilon_{k+1}^{(*)} \geq \max_{k \in \mathbf{k}} \{ \min_{\mathbf{T}_i} \{ \| \partial_{w_k} J_{w_k^*}(\mathbf{T}_i) \| \| dw_k^* \| \tag{LB}$$

$$+ \sum_{T(k) \in \mathbf{T}_i} \sum_{i=k}^{K} \| \partial_{T(k)} E_{t \in T(i)} \ell_{w_i^*}(t) \| \| dT(k) \| \} \},$$

*Proof.* See Appendix C        □

It is straightforward to see that this lower bound in Theorem 1 is zero, given no change in tasks $(dT(k))$ or the weights $(dw_k^{(*)})$. However, in practice each time a task $k$ is introduced to the CL problem, there is a change in the value function. This change is an accumulation of the impact of the new task $k$, on all the prior tasks that in the interval $[1,k]$ ($\sum_{i=1}^{k}$ at the outer of the two terms in (LB) accumulates this change). For each task $i$ in this sum, (LB) is a function of two key terms, (I) "the norm of the gradient of the value function with respect to the solution of the CL problem at $i^{th}$ task" and (II) "the norm of the change in the value function due to change in the data at the $i^{th}$ task." We now study the effects of each of these terms below.

**(I)-Capacity diverges (deteriorates) for bounded weight updates:** To illustrate the effect of weight update, we assume that experience replay (ER)-driven CL methods define either (i) a forgetting cost using all the available tasks, and/or (ii) utilize a regularizer on top of the forgetting cost [9]. We further assume that, at each task $k$ the weights are updated for a total of $I$ steps. Under these assumptions, we show that for both settings (i) and (ii) above, the effective capacity diverges. We now state the following theorem.

**Theorem 2.** *Fix $k \in \mathbb{N}$ and $I$, the number of weight updates required to obtain the optimal value. Assume that $\| \partial_{w_k} J_{w_k^*}(\mathbf{T}_i) \| \geq \Phi_w$, $\| \partial_{T(k)} E_{t \in T(i)} \ell_{w_i^*}(t) \| \geq \Phi_T$, and let the smallest value of $\min_{T(k)} \| dT(k) \| \geq \Phi_{dT}$. Let $L, \mathcal{R}$ be the Lipschitz constants for the cost function and the regularization function respectively with $\alpha_{MIN}$ being the smallest learning rate. Then, $\sum_k^K d\epsilon_k^{(*)}$ diverges as a function of $K$, and $I$ with and without the regularization factor.*

*Proof.* See Appendix D        □

Theorem 2 demonstrates an important and novel result in the CL literature. In essence, for any CL algorithm in the literature with standard gradient driven optimization regime, capacity will diverge as long as the each subsequent tasks keeps accumulating constant albeit small differences. Therefore, CL algorithms have the potential to result in a model that does not represent all the tasks reasonably. Moreover, this behavior is uncontrollable because the tasks are unknown apriori.

**(II)-Capacity diverges (deteriorates) when you have a constant change in the tasks:** To demonstrate the effect of tasks on capacity, we state the following theorem

**Theorem 3.** *Under the condition of Theorem 2, let the maximum change in subsequent tasks and weights be given by $\max_{k \in \mathbf{k}} \{\Phi_T \Phi_{dT}\} = c$. Then, the $\sum_k^K d\epsilon_k^{(*)}$ diverges as a function of $K$, and $I$ without any assumptions on the weight updates.*

*Proof.* See Appendix E ▢

Theorem 3 shows that when a constant change is introduced into the tasks even without any assumptions on the weights, the model becomes unsuitable to represent the tasks. The impact of task similarity on CL has also been studied in [35, 15, 27, 20]. In contrast with Theorem 3, [35, 15, 20] study the impact for a linear classifier. In particular, [20][Theorem 3] shows a monotonic decrease in forgetting cost as a function of similarity. For a two task case, Theorem 3 indicates the same result in [20][Theorem 3] as similar tasks will result in no change in capacity. At first, Theorem 3 might appear contradictory to [27][Theorem 3.7], however, our result actually aligns with [27][Theorem 3.7]. Note that in the case when the overlap between distributions will keep decreasing, the loss function will proportionately increase and the risk gap will diverge.

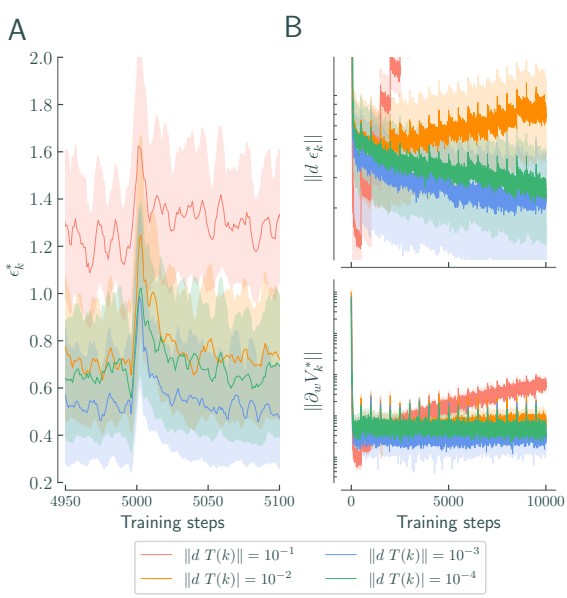

Figure 2: A: Forgetting cost with ER; B: (top) capacity; (bottom) the gradient of capacity with respect to weights as a function of training steps.

### 4.2 EXPERIMENTAL ANALYSIS

In this section, we aim to substantiate the theoretical results and to that end, we develop an array of experiments where we show that capacity diverges with respect to change in tasks irrespective of the type and scale of the model. In all these experiements, we measure the capacity $\epsilon_k^*$, the first difference in capacity $d\,\epsilon_k^*$ and the derivative of the value function with respect to weights, $\partial_w V_k^{(*)}$. We emphasize that, *this work does not present a new method nor does it pertains to demonstrating a new way of doing CL*, but, the goal is to elucidate how the shift in the data-distribution affects the neural network model in the CL setting. To illuminate on this perspective, we build our experiments on popular neural network architectures, namely: feed forward NN (FNN), convolutional NN (CNN), graph NN (GNN) and a transformer-based model. We argue that, for any particular model, the phenomenon of deteriorating capacity as observed on one dataset does translate to other datasets as well because the divergence of capacity is the function of how the NN model react to the shift in the data distribution. Therefore, we choose datasets that are easier to analyze but still relevant in the CL paradigm, both in the supervised and the self-supervised learning regimes. In particular, we utilize a FNN with a synthetic sine wave dataset [25], a CNN with the Omniglot dataset [2], a GNN with synthetic graph dataset and a transformer-driven large language model (LLM) on a trillion (T) tokens dataset provided by RedPajama [8]. We execute FNN/CNN/GNN experiments using the JAX library and we utilize pytorch for the LLM experiments.

**Case Study 1: Feed-forward NNs** *Setup:* For this experiment, we generate a total of twenty tasks, where each task is comprised of sine waves, generated by increasing the value of amplitude and frequency by a quantity $\|dT(k)\|$ to indicate distribution shift. For analysis, we observe the trend of $\epsilon_k^{(*)}$ (capacity) for two standard methods in CL: Experience Replay (ER) shown in Fig. 2 and regularized ER shown in Fig. 3. We simulate four versions of this twenty task CL problem by choosing different values of distribution shift $\|dT(k)\|$, i.e. $\|dT(k)\| \in \{10^{-01}, 10^{-02}, 10^{-03}, 10^{-04}\}$ and

learn twenty tasks for a total of 10 repetitions using mean squared error (MSE) as a cost function and 500 epochs per task.

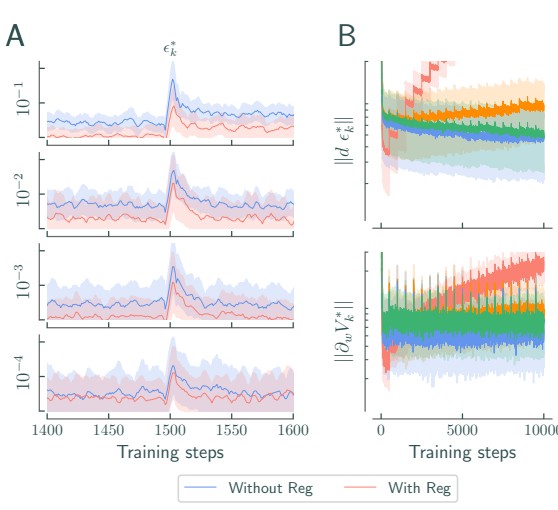

*Analysis of CL using ER:* In panel A of Fig. 2, we plot the mean of capacity, evaluated using its upper bound which is the forgetting cost evaluated using the mean squared error (MSE) averaged across 10 repetitions. The standard deviation is represented using a lighter shaded region.

We first note that, for any new task (we choose a random task at the middle of the learning process to illustrate this), there is an instantaneous increase in the capacity (upper bounded by the forgetting cost). This increase is then minimized by the optimizer, a phenomenon known as stability gap ([24]) in the CL literature. We observe that, the smaller the value of $\|dT(k)\|$, the closer to zero, the capacity appears to be. Our theoretical result in Theorem 3 precisely indicates that each small change in the task leads to a proportional change in the forgetting cost and by extension, the capacity.

Figure 3: A: Forgetting cost with ER and $L_2$ regularization; B: capacity; (bottom) the gradient of capacity with respect to weights as a function of training steps under $L_2$ regularization.

We see this trend also in Fig. 2, Panel B, where we plot $d\epsilon_k^{(*)}$ with respect to training steps. For each new task, the same behavior as Fig. 2, Panel A is observed. Similar to Fig. 2, Panel A, the capacity of the network gets worse proportional to $\|dT(k)\|$ (a conclusion from Theorem 3). As seen in Fig. 2, Panel B, using vanilla experience replay, which is supposed to compensate for the distribution shift in tasks, also exhibits deterioration in capacity. Moreover, the deterioration is proportional to $\|dT(k)\|$ (green is poorer than blue, orange is worse than green, red corresponds to the worst capacity) – an expected result shown in Theorem 2. The addition of a regularization factor does seem to improve this behavior as seen in Fig.3, Panel B. Similarly, Fig.3, Panel A reinforces the observation that regularization applied to ER improves the slope of the capacity for all values of $\|dT(k)\|$ (the different rows in Panel A). As shown in Theorem 2, in spite of regularization, for a large enough $\|dT(k)\|$ the capacity increases drastically (the red curve corresponding to $10^{-01}$ increases very fast) as shown in Fig.3, Panel B.

**Case Study 2: Convolutional NNs**

*Setup:* We now use the Omniglot dataset [9, 2] which is commonly used in continual [2], and meta continual learning problems [25] because of the presence of large numbers of tasks in contrast to the MNIST and CIFAR datasets, that are mostly image recognition datasets. We create a total of 10 classes and sequentially expose the CNN to one class at a time under the incremental class learning paradigm [36] minimizing the cross entropy loss.

*Analysis:* Overall, all the conclusions from the previous case study does carry forward. The stability gap [24] phenomenon is seen in Fig. 4, Panel A. The continuously deteriorating capacity that was observed in Fig. 2 for large noise values are not observed here because, there is no artificial noise being introduced here. In fact, the top plot in Fig. 4, Panel B shows a very stable learning behavior. However, on careful analysis, one can observe that the amount of weight

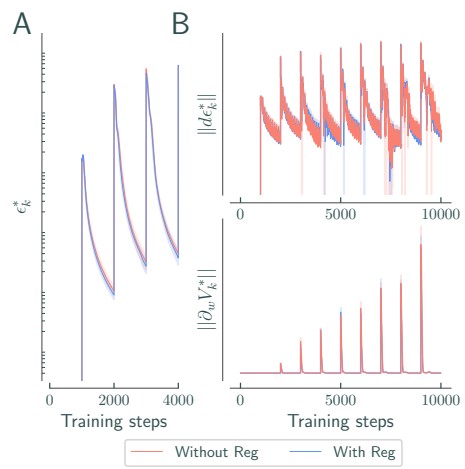

Figure 4: Panel A: (top) Capacity and (bottom) the gradient of capacity; Panel B: Forgetting cost; as a function of training instances.

updates required to attain this learning behavior keeps increasing (bottom plot in Fig. 4, Panel B). This increasing requirement for larger and larger weight updates results in steady deterioration in capacity, as the model is unable to reduce the forgetting cost back to the same level for incoming tasks. This can be observed in Fig. 4, Panel A where capacity at step 2000 is better than that at step 3000 which is better than that at step 4000, and this is also our main contention in Theorems 3 and 2. Although, deteriorating capacity was easier to observe in the synthetic dataset, we show that, even for a real world benchmark CL problem (with no additional noise), the theoretical results are indeed valid.

**Case Study 3: Graph NN**

*Setup:* We generate a total of 10 tasks using the PyTorch geometric library [18] with each task comprising of 4 randomly sampled classes from a 10-class classification problem. The key feature of this synthetic data is that both the node and edge features change. We serially feed these tasks to the graph neural network and train the model for a total of 500 steps each.

*Analysis:* In this study, we again analyze the capacity deterioration from the perspective of data distribution shift due to incoming tasks. In Fig. 5, in the top panel, we show the change in $\|d\epsilon_k^*\|$ (capacity change) for each subsequent tasks. By Lemma (FD), $\|d\epsilon_k^*\|$ is approximately the sum of $\|\frac{\partial V^{(*)}}{\partial w}\|$, $\|\frac{\partial V^{(*)}}{\partial x}\|$ and $\|\frac{\partial V^{(*)}}{\partial \phi}\|$. To observe what introduces the change in capacity, we provide a more granular breakdown of the capacity change by contrasting it with corresponding changes in the

Figure 5: Effect of graph data on the weight updates

input data. We observe that large changes in $\|d\epsilon_k^*\|$, are explained by corresponding large changes in model weights $\frac{\partial V^{(*)}}{\partial w}$ which is directly guided by the change in the tasks $x$ and $\phi$. More specifically, where there is a large spike in the edge or node features (around step 4000), there is a large update in the weights and correspondingly in $\|d\epsilon_k^*\|$ as well. The size of the jump corresponding to weight updates also increases with subsequent tasks.

**Case Study 4: Transformer-based Large Language Models (8M and 134M parameters)**

*Setup:* We utilize four sub-datasets (`wiki` → `git` → `arxiv` → `books`) from the RedPajama 1T tokens dataset [8] for both pre- and continual pre-training. We use the LLama2 tokenizer [47] and decoder model architecture [47] to construct models with 8M and 134M parameters (details in Appendix). Pre-training was done with a batch size of approx. 4M tokens for 48K steps (about 200B tokens), and a 2K-step linear warmup. For CL, we conduct two experiments: one without ER, using data from only the current task, and another with ER, mixing 80% current task data with 20% from previous tasks (details on data mix in Appendix F). Each task is trained for 12K steps (about 50B tokens), starting each new task from the previous task's final checkpoint. Validation scores are computed on the `C4-en` validation set [44] using the final checkpoint for each task. We use identical hyper-parameter settings for both models and leverage PyTorch FSDP [54] on 64 A10 (40GB) GPUs.

*Analysis:* We compare the capacity (measured using its upper bound which is the forgetting cost) for continual pre-training with and without ER of 8M and 134M parameter models in Fig. 6. Pre-training capacity for both models are also shown for reference.

*8M model:* Without ER, we see that the capacity initially goes down for the second task (`git`) but then keeps increasing with the arrival of each new task (`arxiv` followed by `books`). This is an expected baseline result [46] and indicates forgetting. Even with ER, we observe an increase in capacity as new tasks arrive. This is a consequence of Theorem 2, as the model needs to learn concepts from a mix of data from multiple tasks. The only exception occurs for the `books` task, where the cost observed with ER is lower than without ER. We attribute this to initialization bias (i.e., optimal solution from the previous task is a good initialization for the current task). This can also be inferred from Theorem 3, where more similarity in task leads to better learning- an effect that has

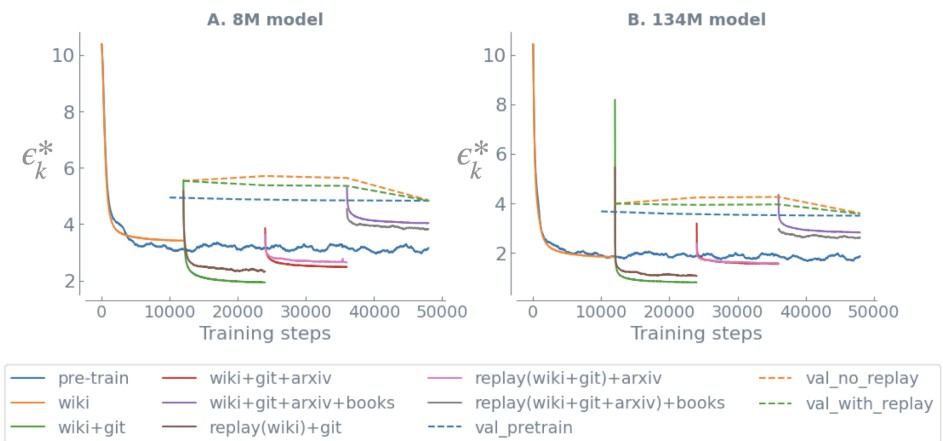

Figure 6: CL on language models demonstrate that forgetting cost increases as new tasks arrive both with and without ER. As expected, the 134M model has higher effective capacity than the 8M model.

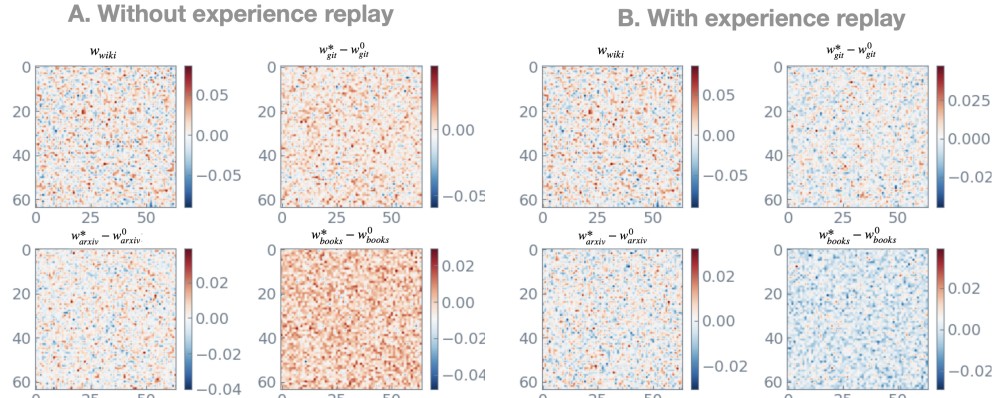

Figure 7: For a task $k$, the $64 \times 64$ heat map shows the difference in weights from the initial value, $\mathbf{w_k^0}$, at the start of training to the final value, $\mathbf{w_k^*}$, at the end of CL training. The weights are randomly sampled from the MLP sublayers in the 8M parameter model. Task arrival order: `wiki` $\rightarrow$ `git` $\rightarrow$ `arxiv` $\rightarrow$ `books`.

been shown theoretically in [35]. The capacity measured on the validation dataset is lower with ER than without it due to improved generalization.

For reference, we add the pre-training capacity curve where all tasks are available together. Initially, the learning objectives (both with and without ER) are relatively easier and therefore task capacity is lower than the overall pre-training capacity. However, as more tasks arrive the capacity eventually becomes higher than the pre-training capacity because the models keeps on forgetting even with ER (Theorem 2). The validation capacity for pre-training model is always lower than both with and without ER indicating that the pre-trained model forgets less than the continual pre-trained model and generalizes better.

*134M model:* We observe very similar behavior as the 8M model, with an increase in capacity as new tasks arrive. However, owing to larger scale, the capacity values are still relatively lower than the 8M model. This is an expected result, as a larger model is more resilient to small changes in the tasks as there are more number of parameters to help with adaptation.

**Case Study 5: Visualization for deeper understanding of the impact of CL on the LLM models**

*Setup:* We randomly sampled $64 \times 64$ parameters (2% of the MLP parameters for 8M and 0.007% for 134M) and tracked how their weights changed from the start ($\mathbf{w_k^0}$) to the end of training ($\mathbf{w_k^*}$) for each task $k$. We then correlated this with the capacity in (Fig. 6). Note, the weight changes caused by each task correspond to the second term in (LB) which is used to characterize capacity. For this

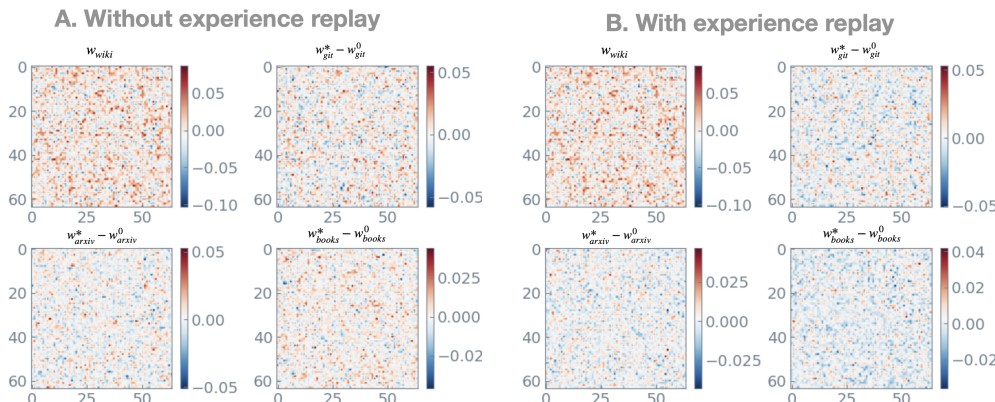

Figure 8: For a task $k$, the $64 \times 64$ heat map shows the difference in weights from the initial value, $\mathbf{w_k^0}$, at the start of training to the final value, $\mathbf{w_k^*}$, at the end of CL training. The weights are randomly sampled from the MLP sublayers in the 134M parameter model. Task arrival order: `wiki` → `git` → `arxiv` → `books`.

example, the last checkpoint from one task serves as the starting point for the next, i.e., $\mathbf{w_k^0} = \mathbf{w_{k-1}^*}$. Although only a small sample of weights was used, repeated trials showed consistent trends.

*Analysis:* For the 8M model without ER, large weight changes (red in Fig. 7(A)) lead to high capacity and increased forgetting. In the `arxiv` task, smaller changes (blue/red) show less learning and more forgetting correlating to the two terms in Lemma 1 where we quantify, how weight and task changes affect the balance point. Significant weight changes occur for the `git` task which effect the second term in Lemma 1 to increase generalization (Fig. 7(A)). In contrast, with ER (Fig. 7(B)), weight changes between tasks are more controlled (more blue than red), reflecting how the two terms in Lemma 1 balance each other. For the `books` task, weight changes are minimal (more blue), indicating marginal model adjustment, lower forgetting, and lower capacity values because the first term no longer balances the second (as shown in Theorem 2).

For the 134M model, we observe similar trends in weight updates. Without ER (Fig. 8(A)), initial changes are slightly larger and continue to increase with each subsequent task. As with the 8M model, increased capacity and significant parameter changes indicate poor representation capability of the model. On the other hand, with ER (Fig. 8(B)), weight changes are more regularized (more blue than red) as prior tasks reduce the amount of increase in the capacity.

## 5   CONCLUSION

We studied capacity in continual learning, focusing on the interplay between the model, tasks, optimization procedure, and their impact on the balance point. We introduce CL's effective model capacity (CLEMC) and find that changes in CLEMC depend on the importance of each task, the cumulative weight changes at each task onset, and the cumulative task changes due to data distribution shifts. Our main conclusion is that even if each subsequent task is only slightly different from the previous one, the effective capacity eventually becomes small, rendering the model unusable.

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

We will begin by restating some preliminaries, these are the exact copy of the initial text in Section 3 of the paper.

# A PRELIMINARIES

Let $x$ and $y$ be random variables corresponding to input and output probability spaces with support $\mathcal{X}$ and $\mathcal{Y}$ and $\mathcal{B}(\mathcal{X})$ and $\mathcal{B}(\mathcal{Y})$ representing the corresponding Borel algebras. Define $t$ as a random variable denoting the joint space of $x \times y$ with a model $f_{(w,h)} : \mathcal{X} \to \mathcal{Y}$ being specified using weights $w$ and hyperparameters $h$. Given compact sets $\mathcal{W}$ over $w$ and $\mathcal{H}$ over $h$, the goal is to learn the weights by searching over the hypothesis space $f = \{f_{(w,h)}, \forall h \in \mathcal{H}, w \in \mathcal{W}\}$ through a loss function $\ell_{w,h}(t)$. In this paper, we will assume that the hyperparameter/architecture is fixed and therefore, will drop the notation $h$ and denote loss simply as $\ell_w(t)$. Throughout the paper, we will assume $x_k = x(k)$ and use them interchangeably, and $\mathbf{k} = [1, 2, \cdots, k]$. In this context, we characterize the effective model capacity as follows.

**Effective Model Capacity:** We will assume that $\ell_w(t)$ is continuous and twice differentiable over the support $\mathcal{X} \times \mathcal{Y}$ or $\mathcal{X}$, and the compact set $\mathcal{W}$. Under these assumptions, let $\ell_{min} = \mathcal{O}_{\mathcal{W}}(T) = min_{w \in \mathcal{W}} \quad E_{t \in T}[\ell_w(t)]$ be the optimization procedure with $T$ being a dataset of samples $t$ with $T \subset \mathcal{B}(\mathcal{T})$. Then, given the best hyperparameter/architecture configurations, the optimization procedure $\mathcal{O}_{\mathcal{W}}$ seeks to find the weights $w^* \in \mathcal{W}$ that minimizes the loss over a dataset. Given this setting, we define the effective model capacity (the upper/lower bounds derived in the appendix) as the smallest achievable loss value using $\mathcal{O}_{\mathcal{W}}$ that remains unchanged even when additional data or training is used.

**Definition 1** [Effective Model Capacity (EMC)] Given $\mathcal{W}$ as the weight space and $T \in \mathcal{B}(\mathcal{T})$ with an optimization procedure $\mathcal{O}_{\mathcal{W}}(T)$, the EMC of the model $f$ is given as

$$\epsilon = \min_{T \in \mathcal{B}(\mathcal{T})} \quad [\mathcal{O}_{\mathcal{W}}(T)] = \min_{T \in \mathcal{B}(\mathcal{T})} \quad [\min_{w \in \mathcal{W}} \quad E_{t \in T}[\ell_w(t)]]$$

Given a feasible weight set $\mathcal{W}_k$, and loss function $\ell_{w_k}(t), t \in \mathcal{T}_k$, the model at $k$ is denoted by $f_{w_k}$, the goal of CL is to maintain memory of all observed tasks, then, the CL forgetting cost for the interval $\mathbf{k} = [1, k]$ is given as

$$\min_{w_k \in \mathcal{W}_k} J_{w_k}(\mathbf{T}_k) = \min_{w_k \in \mathcal{W}_k} \sum_{i=1}^{k} \gamma_i \left[ E_{t \in T(i)} [\ell_{w_k}(t)] \right] , \quad \forall T(i) \in \mathbf{T}_k,$$

where, $\gamma$ ensures boundedness of $J_{w_k}(\mathbf{T}_k)$ (see [45], Lemma 1). For a fixed $h \in \mathcal{H}$, the complete CL problem is

$$V^{(*)}(u_k) = \min_{u_k} \quad \sum_{i=k}^{K} [J_{w_i}(\mathbf{T}_i)], u_k = \{w_i, i = k, k+1, \cdots K\}$$

**CL Effective Model Capacity and Balance Point:** For ease of exposition, we begin by stating

**Definition 2** [Forgetting Effective Model Capacity (FEMC)] For task $k \in [1, K]$, dataset $\mathbf{T}_k$, weight space $\mathcal{W}_k$, optimization procedure $\mathcal{O}_{\mathcal{W}_k}(\mathbf{T}_k)$, EMC at $k$, $\epsilon_k = min_{\mathbf{T}_k, w_k} J_{w_k}(\mathbf{T}_k)$, we define FEMC at task $k$ as:

$$\text{FEMC}(k) = \max_{\mathbf{k}} \epsilon_{\mathbf{k}} = \max\{\epsilon_1, \epsilon_2, \cdots, \epsilon_k\}$$

$FEMC(k)$ at each $k$ is defined by the highest forgetting loss in the interval $[1, k]$. We now define CL effective model capacity as follows.

**Definition 3** [Effective Model Capacity for CL (CLEMC)] For a task $k \in [1, K]$, we define CLEMC as the sum of FEMC across all possible tasks as

$$\epsilon_k^{(*)} = \sum_{i=k}^{K} \text{FEMC}(i) = \sum_{i=k}^{K} \max_{\mathbf{i}} \epsilon_{\mathbf{i}}$$

We will derive the notion of first difference in capacity as a function of the forgetting cost

# B   FIRST DIFFERENCE

**Lemma 1.** For $k \in [1, K]$, let $u_k = \{w_i, i = k, k+1, \cdots K\}$ be weight sequences from $k$ with $\mathcal{U}(k) = \{\mathcal{W}_i, i = k, k+1, \cdots\}$– the compact sets. Next define $(J_F)$, (CL) and (CLEMC) to write

$$\epsilon_{k+1}^{(*)} - \epsilon_k^{(*)} = min_{\mathbf{k}} \{ \max_{\mathbf{T}_i} \{\langle \partial_{w_k} V^{(*)}(u_k), dw_k \rangle + \sum_{T \in \mathbf{T}_k} \langle \partial_T V^{(*)}(u_k), dT \rangle\}\}$$

*Proof.* We first derive the current forgetting cost as a function of infinitesimal change in $V^{(*)}(u_k)$ in the following technical lemma.

**Lemma.** *Consider $k \in [0, K]$ with the forgetting cost as in $(J_F)$ and CL problem in* (CL). *Then,*

$$- \min_{w_k} J_{w_k}(\mathbf{T}_k) = \left\langle \partial_{w_k} V^{(*)}(u_k), dw_k \right\rangle + \sum_{T \in \mathbf{T}_k} \left\langle \partial_T V^{(*)}(u_k), dT \right\rangle + \mathcal{O}(2) \tag{1}$$

*where $d$ is the first difference operator, $\partial$ refers to the first derivative and $\mathcal{O}(2)$ represent the higher order derivative terms.*

*Proof.* Let $u_k = \{w_i, i = k, k+1, \cdots K\}$ be the sequence of weights starting from $k$ with $\mathcal{U}(k) = \{\mathcal{W}_i, i = k, k+1, \cdots\}$ being the sequence of their respective compact sets. Under the assumption that the optimal cost $V^{(*)}(u_k)$ is given by the optimal trajectory of weights $u_k$ corresponding to the tasks sets $\{T_i \in \mathcal{T}_i, i = k, k+1, \cdots K\}$, we can write the following system of recursive equations

$$V^{(*)}(u_k) = \min_{u_k \in \mathcal{U}_k} \sum_{i=k}^{K} [J_{w_i}(\mathbf{T}_i)] \tag{2a}$$

$$V^{(*)}(u_{k+1}) = \min_{u_{k+1} \in \mathcal{U}_{k+1}} \sum_{i=k+1}^{K} [J_{w_i}(\mathbf{T}_i)] \tag{2b}$$

$$V^{(*)}(u_k) = \min_{w_k} J_{w_k}(\mathbf{T}_k) + V^{(*)}(u_{k+1}) \tag{2c}$$

where (2a) and (2b) follow directly from using $(J_F)$ and (2c) is obtained by simply rewriting (2a) using (2b).

Now, given two trajectories $u_k$ and $u_{k+1}$, the change introduced by $u_{k+1}$ to $V^{(*)}(u_k)$ is given by Taylor series approximation of $V^{(*)}(u_k)$ around $w_k$ and $\mathbf{T}_k$ as,

$$V^{(*)}(u_{k+1}) = V^{(*)}(u_k) + \left\langle \partial_{w_k} V^{(*)}(u_k), dw_k \right\rangle + \sum_{T \in \mathbf{T}_k} \left\langle \partial_T V^{(*)}(u_k), dT \right\rangle + \mathcal{O}(2) \tag{3}$$

where $dT_k$ and $dw_k$ are the infinitesimal perturbations to data and weights respectively and $\mathcal{O}(2)$ represent higher order derivative terms. Substituting (3) into (2c) to get

$$\cancel{V^{(*)}(u_k)} = \min_{w_k} J_{w_k}(\mathbf{T}_k) + \cancel{V^{(*)}(u_k)} + \left\langle \partial_{w_k} V^{(*)}(u_k), dw_k \right\rangle + \sum_{T \in \mathbf{T}_k} \left\langle \partial_T V^{(*)}(u_k), dT \right\rangle + \mathcal{O}(2)$$

which proves the result stated in the technical Lemma. $\qquad \square$

Using the above result, we can now prove Lemma (1). Towards this end, we begin by writing,

$$\epsilon_k^{(*)} = \sum_{i=k}^{K} max_{\mathbf{i}} \, \epsilon_{\mathbf{i}} = max_{\mathbf{k}} \, \epsilon_{\mathbf{k}} + \epsilon_{k+1}^{(*)} \tag{4a}$$

$$\epsilon_{k+1}^{(*)} - \epsilon_k^{(*)} = min_{\mathbf{k}} \{ -\epsilon_{\mathbf{k}} \} = min_{\mathbf{k}} \{ -\{\min_{\mathbf{T}_i} \min_{w_i} J_{w_i}(\mathbf{T}_i)\}, \, i \in \mathbf{k} \} \tag{4b}$$

$$= min_{\mathbf{k}} \{ \max_{\mathbf{T}_i} (-\min_{w_i} J_{w_i}(\mathbf{T}_i)), \, i \in \mathbf{k} \} \tag{4c}$$

where (4a) is obtained by applying (CLEMC), and (4b) is obtained by rewriting $\epsilon_{\mathbf{k}}$ using $(J_F)$. Substituting (1) into (4c), and ignoring the higher order derivative terms denoted by $\mathcal{O}(2)$ [3], we obtain the result as

$$\epsilon_{k+1}^{(*)} - \epsilon_k^{(*)} = min_{k \in \mathbf{k}} \{ \max_{\mathbf{T}_i} (-\min_{w_i} J_{w_i}(\mathbf{T}_i)), i \in \mathbf{k} \} \tag{5a}$$

$$= min_{k \in \mathbf{k}} \{ \max_{\mathbf{T}_i} \{ \left\langle \partial_{w_k} V^{(*)}(u_k), dw_k \right\rangle + \sum_{T \in \mathbf{T}_k} \left\langle \partial_T V^{(*)}(u_k), dT \right\rangle \} \tag{5b}$$

$$\square$$

Next, we will derive the lower bound on the first difference in capacity which stems from a lower and upperbound on capacity.

## C   LOWER BOUND ON FIRST DIFFERENCE

**Theorem 1.** The first difference in CLEMC (FD) is lower bounded as

$$\epsilon_k^{(*)} - \epsilon_{k+1}^{(*)} \geq \max_{k \in \mathbf{k}} \{ \min_{\mathbf{T}_i} \{ \| \partial_{w_k} J_{w_k^*}(\mathbf{T}_i) \| \| dw_k^* \|$$

$$+ \sum_{T(k) \in \mathbf{T}_i} \sum_{i=k}^{K} \| \partial_{T(k)} E_{t \in T(i)} \ell_{w_i^*}(t) \| \| dT(k) \| \} \},$$

*Proof.* From Lemma (1) we get

$$\epsilon_{k+1}^{(*)} - \epsilon_k^{(*)} = min_{k \in \mathbf{k}} \{ \max_{\mathbf{T}_k} \{ \left\langle \partial_{w_k} V^{(*)}(u_k), dw_k \right\rangle + \sum_{T \in \mathbf{T}_k} \left\langle \partial_T V^{(*)}(u_k), dT \right\rangle \} \}$$

$$\leq min_{k \in \mathbf{k}} \{ \max_{\mathbf{T}_k} \{ \| \partial_{w_k} V^{(*)}(u_k) \| \| dw_k \| + \sum_{T \in \mathbf{T}_k} \| \partial_T V^{(*)}(u_k) \| \| dT \| \} \} \tag{6a}$$

where (6a) is obtained using Cauchy-Schwarz inequality, $\langle a, b \rangle \leq \|a\| \|b\|$. We then bound both the gradient norm terms in (6a) as follows.

For the first gradient norm term, we assume that the optimal cost, $V^{(*)}$, is given by the weight trajectory $u_k$ with $u_k = \{ w_i, i = k, k+1, \cdots K \}$. We can then bound it through the following inequalities.

$$\| \partial_{w_k} V^{(*)}(u_k) \| = \| \partial_{w_k} \min_{u_k} \sum_{i=k}^{K} J_{w_i}(\mathbf{T}_i) \| \tag{7a}$$

$$\leq \| \partial_{w_k} \sum_{i=k}^{K} min_{w_i} J_{w_i}(\mathbf{T}_i) \| \tag{7b}$$

$$\leq \| \sum_{i=k}^{K} \partial_{w_k} min_{w_i} J_{w_i}(\mathbf{T}_i) \| \tag{7c}$$

$$\leq \| \partial_{w_k} min_{w_k} J_{w_k}(\mathbf{T}_i) \| \tag{7d}$$

where (7b) is because the norm of the gradient, with respect to weights, at the optimal cost (due to an optimal trajectory) is always less than the norm of the gradient, with respect to the weights, at a forgetting cost corresponding to any arbitrary weight trajectory. (7c) follows from the sum rule of derivatives and (7d) is because all terms from $w_{k+1}$ onwards vanish due to lack of dependence on $w_k$.

For the second norm of the gradient term in (6a), we again write the optimal cost $V^{(*)}(u_k) = \sum_{i=k}^{K} min_{w_i} J_{w_i}(\mathbf{T}_i)$ such that $\mathbf{T}_i = \{ T(1), \cdots T(i) \}$. We further observe that if the optimal cost is differentiated with respect to $T(k)$ only the $k^{th}$ term in the inner sum will remain. We can then bound it through the following inequalities.

$$\|\partial_{T(k)} V^{(*)}(u_k)\| \leq \ \|\partial_{T(k)} \sum_{i=k}^{K} min_{w_i} J_{w_i}(\mathbf{T}_i)\| \tag{8a}$$

$$\leq \ \|\sum_{i=k}^{K} \partial_{T(k)} min_{w_i} \sum_{p=1}^{i} E_{t \in T(p)} \ell_{w_i}(t)\| \tag{8b}$$

$$\leq \ \|\sum_{i=k}^{K} \partial_{T(k)} E_{t \in T(i)} \ell_{w^*(i)}(t)\| \tag{8c}$$

Then, upon substituting (7d) and (8c) into (6a) we get,

$$\epsilon_{k+1}^{(*)} - \epsilon_k^{(*)} \leq \min_{k \in \mathbf{k}} \{ \max_{\mathbf{T}_i} \{ \|\partial_{w_k} J_{w_k^*}(\mathbf{T}_i)\| \|dw_k^*\| + \sum_{T(k) \in \mathbf{T}_i} \sum_{i=k}^{K} \|\partial_{T(k)} E_{t \in T(i)} \ell_{w_i^*}(t)\| \|dT(k)\| \} \}, \tag{9}$$

where we have replaced the inner minimization problem with respect to weights by the corresponding $w^*$. Multiplication with $-1$ provides the lower bound as

$$\epsilon_k^{(*)} - \epsilon_{k+1}^{(*)} \geq \max_{k \in \mathbf{k}} \{ \min_{\mathbf{T}_i} \{ \|\partial_{w_k} J_{w_k^*}(\mathbf{T}_i)\| \|dw_k^*\| + \sum_{T(k) \in \mathbf{T}_i} \sum_{i=k}^{K} \|\partial_{T(k)} E_{t \in T(i)} \ell_{w_i^*}(t)\| \|dT(k)\| \} \}, \tag{10}$$

$\square$

This lower bound then leads to the conclusion that capacity is non-stationary and diverges with increase in weight update or divergence between subsequent tasks. This non-stationarity extends to experience replay and experience replay with regularization

# D  DIVERGENCE WITH RESPECT TO WEIGHTS

**Theorem 2.** Fix $k \in \mathbb{N}$ and $I$, the number of weight updates required to obtain the optimal value. Assume that $\|\partial_{w_k} J_{w_k^*}(\mathbf{T}_i)\| \geq \Phi_w$, $\|\partial_{T(k)} E_{t \in T(i)} \ell_{w^*}(t)\| \geq \Phi_T$, and let the smallest value of $min_{T(k)} \|dT(k)\| \geq \Phi_{dT}$. Let $L, \mathcal{R}$ be the Lipschitz constants for the cost function and the regularization function respectively with $\alpha_{\text{MIN}}$ being the smallest learning rate. Then, $\sum_k^K d\epsilon_k^{(*)}$ diverges as a function of $K$, and $I$ with and without the regularization factor.

*Proof.* We first prove the technical Lemma below.

**Lemma.** *Fix $k \in \mathbb{N}$ and let the weights at any task $k$ be updated for a total of $I$ steps. Assume $T(k)$ is provided through a series of batches such that $T(k) = \{t_k^{(i)}, i = 1, \cdots, I\}$ with $t_k^{(i)}$ be a tensor corresponding to batch of data at the $i^{th}$ step for the $k^{th}$ task, sampled uniformly from the underlying support. For the $i^{th}$ update step of the $k^{th}$ task, let the forgetting cost be denoted by $J_{w_k}(\mathbf{T}_k)$, gradient be denoted by $g_k^{(i)}$, and learning rate by $\alpha_k^{(i)}$. Then,*

$$dw_k^* = -\sum_{i=0}^{I-1} \alpha_k^{(i)} g_k^{(i)} \tag{11}$$

*Proof.* Note now that, we abuse notation to define $dw_k^* = w_k^* - w_k^{(0)} = w_k^{(I)} - w_k^{(0)}$ assuming that the optimal point is achieved after $I$ updates (indicated by parenthesis). Then, at any particular update step, we obtain

$$w_k^{(i+1)} = w_k^{(i)} - \alpha_k^{(i)} g_k^{(i)} \tag{12}$$

where $g_k^{(i)}$ is the update gradient at the this step.

$$w_k^{(i+1)} = w_k^{(i)} - \alpha_k^{(i)} g_k^{(i)} \tag{13}$$

We may now write the sum over the I steps at a

$$w_k^{(1)} = w_k^{(0)} - \alpha_k^{(0)} g_k^{(0)} \tag{14a}$$

$$w_k^{(2)} = w_k^{(1)} - \alpha_k^{(1)} g_k^{(1)} \tag{14b}$$

$$\vdots \tag{14c}$$

$$w_k^{(I)} = w_k^{(I-1)} - \alpha_k^{(I-1)} g_k^{(I-1)} \tag{14d}$$

Adding all these terms to write

$$dw_k^* = -\sum_{i=0}^{I-1} \alpha_k^{(i)} g_k^{(i)} \tag{15}$$

$\square$

Given the first difference in capacity from the technical Lemma above, and under the assumption that $\|\partial_{w_k} J_{w_k^*}(\mathbf{T}_i)\| \geq \Phi_w$ and $\|\partial_{T(k)} E_{t \in T(i)} \ell_{w_i^*}(t)\| \geq \Phi_T$

$$\epsilon_k^{(*)} - \epsilon_{k+1}^{(*)} \geq \max_{k \in \mathbf{k}} \{ \min_{\mathbf{T}_i} \{ \|\partial_{w_k} J_{w_k^*}(\mathbf{T}_i)\| \|dw_k^*\| + \sum_{T(k) \in \mathbf{T}_i} \sum_{i=k}^{K} \|\partial_{T(k)} E_{t \in T(i)} \ell_{w_i^*}(t)\| \|dT(k)\| \} \}$$

$$\geq \max_{k \in \mathbf{k}} \{ \min_{\mathbf{T}_i} \{ \Phi_w \|dw_k^*\| + \sum_{T(k) \in \mathbf{T}_i} \sum_{i=k}^{K} \Phi_T \|dT(k)\| \} \} \tag{16a}$$

$$\geq \max_{k \in \mathbf{k}} \{ \Phi_w \|dw_k^*\| + \min_{\mathbf{T}_i} \sum_{T(k) \in \mathbf{T}_i} \sum_{i=k}^{K} \Phi_T \|dT(k)\| \} \} \tag{16b}$$

$$\geq \max_{k \in \mathbf{k}} \{ \Phi_w \|dw_k^*\| + \sum_{T(k) \in \mathbf{T}_i} \sum_{i=k}^{K} \Phi_T \min_{T(k)} \|dT(k)\| \} \} \tag{16c}$$

Let the smallest value of $\min_{T(k)} \|dT(k)\| \geq \Phi_{dT}$, then, we can write

$$\epsilon_k^{(*)} - \epsilon_{k+1}^{(*)} \geq \max_{k \in \mathbf{k}} \{ \Phi_w \|dw_k^*\| + \sum_{T(k) \in \mathbf{T}_i} \sum_{i=k}^{K} \Phi_T \Phi_{dT} \} \tag{17}$$

$$\geq \max_{k \in \mathbf{k}} \{ \Phi_w \|dw_k^*\| \} + \max_{k \in \mathbf{k}} \{ \sum_{T(k) \in \mathbf{T}_i} \sum_{i=k}^{K} \Phi_T \Phi_{dT} \} \tag{18}$$

Taking sum from $k$ to K provides with the fact that each $\mathbf{T}_k$ has a total of $k$ sub datasets.

$$\epsilon_k^{(*)} - \epsilon_K^{(*)} \geq \sum_{k}^{K} \left[ \max_{k \in \mathbf{k}} \{ \Phi_w \|dw_k^*\| \} + \max_{k \in \mathbf{k}} \{ \sum_{T(k) \in \mathbf{T}_k} \sum_{i=k}^{K} \Phi_T \Phi_{dT} \} \right] \tag{19}$$

$$\geq \sum_{k}^{K} \max_{k \in \mathbf{k}} \{ \Phi_w \|dw_k^*\| \} + \sum_{k}^{K} \max_{k \in \mathbf{k}} \{ \sum_{T(k) \in \mathbf{T}_k} (K-k) \Phi_T \Phi_{dT} \} \tag{20}$$

$$\geq \sum_{k}^{K} \max_{k \in \mathbf{k}} \{ \Phi_w \|dw_k^*\| \} + k(K-k)^2 \max_{k \in \mathbf{k}} \{ \Phi_T \Phi_{dT} \} \tag{21}$$

Since, $\max_{k \in \mathbf{k}} \{ \Phi_T \Phi_{dT} \} = \Phi_T \Phi_{dT}$, $\max_{k \in \mathbf{k}} \{ \Phi_w \Phi_{dw} \} = \Phi_w \Phi_{dw}$ and $\|dw_k^*\| \geq \Phi_{dw}$, we write

$$\epsilon_k^{(*)} - \epsilon_K^{(*)} \geq \sum_{k}^{K} \max_{k \in \mathbf{k}} \{ \Phi_w \Phi_{dw} \} + k(K-k)^2 \Phi_T \Phi_{dT} \geq \sum_{k}^{K} \Phi_w \Phi_{dw} + k(K-k)^2 \Phi_T \Phi_{dT}$$

$$\tag{22a}$$

We will now assume that the changes introduced by the task are bounded over all future and past tasks. Given that $K > 0, k > 0, \Phi_w > 0, \Phi_T > 0, \Phi_{dT} > c$, we obtain

$$\epsilon_k^{(*)} - \epsilon_K^{(*)} \geq (K-k)\Phi_w\Phi_{dw} + k(K-k)^2 \Phi_T c \tag{23}$$

Now, by assumption that, for each task, the optimal value of weight is obtained after updating the weights for a total of $I$ steps provides $\Phi_{dw} \geq -\sum_{i=0}^{I-1}\alpha_k^{(i)}g_k^{(i)} \geq -\sum_{i=0}^{I-1}\alpha_k^{(i)}(-L) \geq \sum_{i=0}^{I-1}\alpha_{\text{MIN}}L \geq I\alpha_{\text{MIN}}L$. Thus, we obtain

$$\epsilon_k^{(*)} - \epsilon_K^{(*)} \geq (K-k)\Phi_w I\alpha_{\text{MIN}}L + k(K-k)^2 \Phi_T c \tag{24}$$

Then $\epsilon_k^{(*)} - \epsilon_K^{(*)}$ diverges as a function of $K, k, I, c$.

Similarly, for the case with regularization we may write $d\Phi_{dw} \geq -\sum_{i=0}^{I}\alpha_k^{(i)}g_k^{(i)} \geq -\sum_{i=0}^{I-1}\alpha_k^{(i)} - (L+\beta\mathcal{R}) \geq \sum_{i=0}^{I-1}\alpha_{\text{MIN}}(L+\beta\mathcal{R}) \geq I\alpha_{\text{MIN}}(L+\beta\mathcal{R})$, where $L, \mathcal{R}$ are the Lipschitz bounds on the gradients and regularizer function respectively and $\beta > 0$ is a coefficient. Thus, we obtain

$$\epsilon_k^{(*)} - \epsilon_K^{(*)} \geq (K-k)\Phi_w I\alpha_{\text{MIN}}(L+\beta\mathcal{R}) + k(K-k)^2 \Phi_T c \tag{25}$$

and we observe divergence as a function of $K, k$. $\qquad\square$

Finally we show our main result, that is, if a small change is introduced by every task, it accumulate to result in a divergent capacity.

# E    DIVERGENCE WITH RESPECT TO TASKS

**Theorem 3.** Under the condition of Theorem 2, let the maximum change in subsequent tasks and weights be given by $\max_{k\in\mathbf{k}} \{\Phi_T\Phi_{dT}\} = c$. Then, the $\sum_k^K d\epsilon_k^{(*)}$ diverges as a function of $K$, and $I$ without any assumptions on the weight updates.

*Proof.* Given the first difference in capacity, and under the assumption that $\|\partial_{w_k}J_{w_k^*}(\mathbf{T}_i)\| \geq \Phi_w$ and $\|\partial_{T(k)}E_{t\in T(i)}\ell_{w_i^*}(t)\| \geq \Phi_T$

$$\epsilon_k^{(*)} - \epsilon_{k+1}^{(*)} \geq \max_{k\in\mathbf{k}} \{\min_{\mathbf{T}_i} \{\|\partial_{w_k}J_{w_k^*}(\mathbf{T}_i)\|\|dw_k^*\| + \sum_{T(k)\in\mathbf{T}_i}\sum_{i=k}^{K}\|\partial_{T(k)}E_{t\in T(i)}\ell_{w_i^*}(t)\|\|dT(k)\|\}\}$$

$$\geq \max_{k\in\mathbf{k}} \{\min_{\mathbf{T}_i} \{\Phi_w\|dw_k^*\| + \sum_{T(k)\in\mathbf{T}_i}\sum_{i=k}^{K}\Phi_T\|dT(k)\|\}\} \tag{26a}$$

$$\geq \max_{k\in\mathbf{k}} \{\Phi_w\|dw_k^*\| + \min_{\mathbf{T}_i}\sum_{T(k)\in\mathbf{T}_i}\sum_{i=k}^{K}\Phi_T\|dT(k)\|\}\} \tag{26b}$$

$$\geq \max_{k\in\mathbf{k}} \{\Phi_w\|dw_k^*\| + \sum_{T(k)\in\mathbf{T}_i}\sum_{i=k}^{K}\Phi_T\min_{T(k)}\|dT(k)\|\}\} \tag{26c}$$

Let the smallest value of $\min_{T(k)}\|dT(k)\| \geq \Phi_{dT}$, then, we can write

$$\epsilon_k^{(*)} - \epsilon_{k+1}^{(*)} \geq \max_{k\in\mathbf{k}} \{\Phi_w\|dw_k^*\| + \sum_{T(k)\in\mathbf{T}_i}\sum_{i=k}^{K}\Phi_T\Phi_{dT}\}$$

$$\geq \max_{k\in\mathbf{k}} \{\Phi_w\|dw_k^*\|\} + \max_{k\in\mathbf{k}} \{\sum_{T(k)\in\mathbf{T}_i}\sum_{i=k}^{K}\Phi_T\Phi_{dT}\} \tag{27}$$

Taking sum from $k$ to K provides with the fact that each $\mathbf{T}_k$ has a total of $k$ sub datasets.

$$\epsilon_k^{(*)} - \epsilon_K^{(*)} \geq \sum_k^K \left[ \max_{k \in \mathbf{k}} \{\Phi_w \|dw_k^*\|\} + \max_{k \in \mathbf{k}} \{ \sum_{T(k) \in \mathbf{T}_k} \sum_{i=k}^K \Phi_T \Phi_{dT} \} \right] \tag{28a}$$

$$\geq \sum_k^K \max_{k \in \mathbf{k}} \{\Phi_w \|dw_k^*\|\} + \sum_k^K \max_{k \in \mathbf{k}} \{ \sum_{T(k) \in \mathbf{T}_k} (K - k)\Phi_T \Phi_{dT} \} \tag{28b}$$

$$\geq \sum_k^K \max_{k \in \mathbf{k}} \{\Phi_w \|dw_k^*\|\} + k(K - k)^2 \max_{k \in \mathbf{k}} \{\Phi_T \Phi_{dT}\} \tag{28c}$$

Since, $\max_{k \in \mathbf{k}} \{\Phi_T \Phi_{dT}\} = \Phi_T \Phi_{dT}$, $\max_{k \in \mathbf{k}} \{\Phi_w \Phi_{dw}\} = \Phi_w \Phi_{dw}$ and $\|dw_k^*\| \geq \Phi_{dw}$, we write

$$\epsilon_k^{(*)} - \epsilon_K^{(*)} \geq \sum_k^K \max_{k \in \mathbf{k}} \{\Phi_w \Phi_{dw}\} + k(K - k)^2 \, \Phi_T \Phi_{dT} \tag{29a}$$

Assuming that the changes introduced by the task are bounded over all future and past tasks, i.e., $\Phi_{dT} > c$, we get

$$\epsilon_k^{(*)} - \epsilon_K^{(*)} \geq \sum_k^K \Phi_w \Phi_{dw} + k(K - k)^2 \Phi_T \, c \tag{30a}$$

Even for a constant change in task, $\epsilon_k^{(*)} - \epsilon_K^{(*)}$ diverges as a function of $K$. $\qquad\square$

# F   DETAILS FOR CASE STUDY 4

We used the following configuration of a transformer block to instantiate the 8M model.

Embedding layer: (32000, 128); Attention layer: (k, q, v, o): (128, 128); MLP layer: gate_projection (128, 256), up_projection (128, 256), down_projection (256, 128); Activation function: SiLU; Layernorm: RMSNorm ; Head layer: (128, 32000); Attention heads: 2 ; Layers: 2 Hidden size: 128.

We used the following configuration of a transformer block to instantiate the 134M model.

Embedding layer: (32000, 768); Attention layer: (k, q, v, o): (768, 768); MLP layer: gate_projection (768, 2048), up_projection (768, 2048), down_projection (2048, 768); Activation function: SiLU; Layernorm: RMSNorm ; Head layer: (768, 32000); Attention heads: 12 ; Layers: 12 Hidden size: 768.

**Pre-training Data Mix:**

- `wiki`: 0.28
- `git`: 0.28
- `arxiv`: 0.16
- `books`: 0.28

**Experience Replay - Data Mix:**

- `wiki`: 1.0
- `wiki`: 0.2, `git`: 0.8
- `wiki`: 0.1, `git`: 0.1, `arxiv`: 0.8
- `wiki`: 0.06, `git`: 0.07, `arxiv`: 0.07, `books`: 0.8

