# OpenReview forum: "On Understanding of the Dynamics of Model Capacity in Continual Learning"
_ICLR.cc/2025/Conference — Submitted to ICLR 2025_

### Official Review · Reviewer_jrow · 2024-10-31

**Soundness:** 2
**Presentation:** 2
**Contribution:** 2
**Rating:** 6
**Confidence:** 3

**Summary:**

The paper proposes Effective Model Capacity for Continual Learning (CLEMC) to quantify and understand how a neural network's capacity evolves in a continual learning setting. It provides theoretical justifications demonstrating that network capacity decreases with even slight changes in task distribution, eventually making the network unusable. The paper includes experiments on diverse network backbones and data modalities to evaluate the claimed theoretical results.

**Strengths:**

1. The paper presents rigorous theoretical justifications and detailed proofs; however, my current expertise limits my ability to fully validate these derivations.

2. The paper attempts to experiment on across different datasets and different modalities.

**Weaknesses:**

1. The paper concludes that with slight changes in tasks, the model's effective capacity decreases, eventually rendering it unusable. While the reduction in capacity appears justified, the paper lacks experiments that clarify what "unusable" entails. Establishing a threshold, perhaps based on loss or accuracy, to define when the model becomes unusable would be beneficial.

2. While the paper targets to justify the theoretical results drawn to four different modalities, the experimental setup vary without any rationale. The experiments in graph NN consists of gradients norm changes, while missing out the capacity change experiments in other modalities. And, the paper is hard to follow at times. For instance, a detailed explanation on figure 1 is missing.

3. The visualizations are hard to analyze. There is no indication of task switching in the figures. A grid line for each task change can help improve the visualizations for both panels in fig 2A, B. The legend for fig 2 panel A conflicts with that of fig 1. So it would be better to have a consistent notation for all results. Also, a marking to indicate 4000 step, mentioned in line 373, would be better to include.

**Questions:**

1. Is $\| \Delta T(p) \|$ substantial compared to the initial values as the values are relatively small? What would be the percentage change compared to initial amplitude and frequency values?
1. Why does the experimental setup vary across modalities? A standardized set of experiments applied consistently across all modalities would enhance the strength of the paper.
2. How does the experiment with regularization having improved forgetting results connect with the theoretical insights provided?
3. What is the rationale behind difference in pattern of Fig 4A and fig 2B? 4A has relatively increasing trends for the spikes while the previous ones don't.
4. How is $\gamma^{(k)}$ selected? Also, additional information on the hyper-parameters used would be appreciated.

---

> ### Author Response · Authors · 2024-11-23
> **Thank you for your review**
>
> **Slight changes in tasks,** CL Effective Model Capacity (CLEMC) is defined in terms of forgetting cost which in turn is defined in terms of the loss. When capacity increases it is actually an increase in the loss. An increased loss is usually indicative of degradation in model performance and that is what we demonstrate using our experiments, without actually quantifying a usability threshold. Furthermore, the loss threshold after which a model becomes unusable is very task-dependent and coming up with a general threshold is difficult.
>
> One example is Fig. 6, where we also plot the validation cost. This indicates the generalization performance of the model. Lower the validation cost better the model. We show that increase in capacity leads to increased validation cost - again indicative of degraded model performance.
>
> **Four Modalities** The experiments are to validate various aspects of the theory. The common theme in each experiment is that we define tasks by changing the data distribution.  Depending on the modality of the experiment, the features we use to change the data distribution varies. In some experiments, we demonstrate the change in capacity with introduction of new tasks by directly plotting $\epsilon^*_k$ or the forgetting cost $J_F$. The choice of using the gradient of the value function w.r.t the weights was to show the forgetting is associated with large changes in these gradient values. This is in line with the result in Theorem 2.
>
> **Detailed explanation on figure 1** We have performed a major revision of the text to make the presentation simple and easy to understand. For instance, we have added a detailed caption explaining Fig. 1 We have also included text describing parts of Fig. 1 in various subsections of Section 3.
> **Hard visualizations** We have made the subsequent changes to the plots for additional clarification.
> **|ΔT(p)|** For each new task, we introduce a $\delta$ change in the amplitude and frequency. For K subsequent tasks, the change is $K \times \delta.$
>
> **Improved forgetting results**  In Theorem 3, we show that capacity divergence happens with change in task distribution even with regularization. In our experiments, we show that even runs with regularization show an increasing trend in capacity (higher forgetting cost) as more tasks arrive. However, note that we have not explicitly analyzed the effect of regularization on the accuracy in this paper. This could be a potential future work.
>
> **Fig 4A and fig 2B**  This is an expected behavior, where for each particular tasks, the model keeps trying to learn and reduces the forgetting cost. The main theoretical and experimental conclusion of this work is that, in spite of such learning, the task behavior eventually becomes too much for the network to handle, thus resulting in diverging capacity.
>
> **How is γ(k)**  For our experiments we chose $\gamma^{k}$ as 1. The hyper-parameters used are specified in the  setup for each experiment. We have also added Appendix F with parameter details for case study 4.  If you need information about a specific setting please let us know and we will be happy to provide more details.
>
>
> Thank you for your comments, please let us know if any of these comments need additonal clarification, we are happy to elaborate.

---

> > ### Comment · Reviewer_jrow · 2024-11-26
> >
> > I appreciate the authors’ reply to the questions and the weaknesses I raised. I understand that setting a quantifying threshold for usability is task dependent and appreciate the clarification regarding the case with regularization and selection of gamma(k) and other hyper-parameters.
> >
> > However, I feel some of my concerns have not been fully addressed.
> > I understand that ek is used in some and gradient of value function in others. My concern is why not use a common set of experiments across all modalities.
> > I understand the theoretical insights align with the experimental results from fig 4A and 2B. But my original question was to know the rationale behind the difference in the patterns of 4A and 2B.  Could you explain this more clearly?

---

> ### Author Response · Authors · 2024-11-30
> **Response**
>
> **However, I feel some of my concerns have not been fully addressed. I understand that ek is used in some and gradient of value function in others. My concern is why not use a common set of experiments across all modalities.**
>
> We apologize for the unsatisfactory response.
>
> We have now updated all the figures to consistently plot $\epsilon^*_k$ across Figs. 2-6. To provide additional insight, we plot
> $ \epsilon^*_k , d \epsilon^*_k $ and $ \partial_w V_k^*  $  In Fig. 2, 3 and 4.
>
> Moreover, we plot $ \partial_x  V_k^* ,  \partial_{\phi} V_k^*$  and $ \partial_x V_k^* $ in Fig, 5 and visualize how weights change for the transformer model. We hope that these plots provide additional clarity and introduce uniformity to our experiments. \
>
> To provide more rationale behind these choices, note that, we performed four experiments with different model architectures and data \
> (1) FNN with Sine wave data, \
> (2) CNN with image data, \
> (3) GNN with graph data and \
> (4) LLM with text data.
>
> In each experiment, we measured the capacity through the forgetting cost, gradient of the value function with respect to weights $\frac{ \partial V^* }{\partial w}, and $ gradient of the value function with respect to input data. In case of graph data, which consists of both node and edge features, the gradient of the value function with respect to input data has two components: \
> - one is w.r.t node features, $\frac{ \partial V^* }{\partial x}$
> - another is w.r.t. edge features, $ \frac{ \partial V^* }{\partial \phi}.$
>
> Now, in each experiment, our goal was to highlight the capacity degradation due to distribution shift introduced by incoming tasks and also validate our theoretical results. For experiments (1), (2) and (4) [Figs. 2, 3, 4, 6], we used $\epsilon^*_k$ through $J_F$ because  $\epsilon_k^*$ is upper bounded by the forgetting cost $J_F.$
>
> For experiment (3) [Fig. 5], in addition to demonstrating that the capacity deteriorates with an increase in $|| d \epsilon^*_k ||$, we perform a fine-grained analysis and breakdown this increase in terms of the sum of the gradient of value function w.r.t weights ($\frac{ \partial V^* }{\partial w}$),  and the gradient of the value function w.r.t data ($\frac{ \partial V^* }{\partial \phi}$, and $\frac{ \partial V^* }{\partial x}$ ) (as derived in Lemma 1). The intention was to provide different perspectives of the same capacity degradation problem.
>
> **I understand the theoretical insights align with the experimental results from fig 4A and 2B. But my original question was to know the rationale behind the difference in the patterns of 4A and 2B. Could you explain this more clearly?**
> There are two aspects to our experiments, first in Experiment 1, we introduce different degree of noise to see how the capacity reacts to different shifts in the distribution, this is shown in Fig. 2B. Next, in Fig 4A, 5 and 6 we use standard datasets and neural network architectures to elucidate, how capacity changes in typical CL problems without introducing drastic noise values. In Fig. 4A, the change in capacity is not drastic as in Fig. 2. Nonetheless, there are two important observations.
>
> - First, $\frac{ \partial V^* }{\partial w}$ increases with each new task.
> - This change influence the place at which the capacity reaches after repeated learning, as observed, the end point of capacity at 2000 is slightly smaller than 3000 and so on.
>
> In other words, the model fails to converge to the same precision with each new task. The differnce between Fig. 4A and 2 is that is "how drastic the effect looks and how close to zero the model gets after repeated updates?" This key difference is introduced because of the cross entropy loss function in Fig. 4, where the model learn much faster than the case with MSE in Fig. 2 and therefore reaches much closer the zero than the case of MSE. Thus the effect of distribution shift is more prominent in Fig. 2 than Fig. 4. The only indication in Fig. 4 is the jumps in the gradient of the value function with respect to the weights.

---

> ### Author Response · Authors · 2024-12-02
> **Response**
>
> We hope that these responses were satisfactory and you would consider revising your scores for them.
>
> Please let us if there is any additional clarification, you might need.

---

> > ### Comment · Reviewer_jrow · 2024-12-02
> >
> > I would like to thank the authors for their replies. Given that they have addressed my concerns now, I would like to raise the score to 6. And, I would highly encourage the authors to include uniformity in experiments in the eventually accepted version. Thank you!

---

> ### Author Response · Authors · 2024-12-03
> **Thank you**
>
> Will do. We thank you for the reviews and for your time !!

---

### Official Review · Reviewer_rQ9K · 2024-11-04

**Soundness:** 3
**Presentation:** 3
**Contribution:** 3
**Rating:** 5
**Confidence:** 3

**Summary:**

This paper explores the stability-plasticity balance in continual learning (CL) from a dynamical systems perspective. To examine the interplay between model, data, and optimization, the authors introduce CLEMC (CL’s Effective Model Capacity), finding that the network eventually becomes ineffective for representing tasks if each new task varies even slightly from the previous one.

**Strengths:**

1) Theoretical justification is provided for the proposed perspective.

2) Extensive experiments is conducted across various architectures, from small feed-forward (FNN) and convolutional networks (CNN) to medium-sized graph neural networks (GNN) and large transformer-based language models (LLMs).

3) They investigate the interplay among model, task, and optimization, a trio previously examined only in pairwise combinations—either model and optimization or model and data, which I think is interesting to understand their complex relationships.

**Weaknesses:**

1) The paper misses a key theoretical study on continual learning that explores catastrophic forgetting and task similarity[1].

2) Could the authors discuss how CLEMC handles task order? I'm particularly interested in understanding CLEMC's sensitivity to the sequence of tasks.

3) Could the authors provide a more detailed and clear explanation of Figure 1? I find it difficult to understand.

4) In Figure 6, for the books task, the authors note that ER (Experience Replay) results in a lower cost compared to the scenario without ER. They attribute this to initialization bias and task similarity. However, the explanation is unclear to me. Could the authors elaborate on this and clarify the type of task similarity measure they are referring to?

[1] Doan, Thang, et al. "A theoretical analysis of catastrophic forgetting through the ntk overlap matrix." International Conference on Artificial Intelligence and Statistics. PMLR, 2021.

**Questions:**

Check the weaknesses.

---

> ### Author Response · Authors · 2024-11-23
> **Response**
>
> **Key theoretical study**   We thank the reviewer for pointing us to the reference. We have included it in the related work section in the revised manuscript. As mentioned [1] explores catastrophic forgetting (CF) in the presence of task similarity..
>
> **Task Ordering**   The formulation of CLEMC does not consider any particular ordering of the tasks. To obtain Theorem 2 and Theorem 3, we only assume that the changes introduced by any task are bounded. In our experiments, especially the one with LLMs (8M and 134M parameter models) we did, however, observe that forgetting cost depended on the ordering of the tasks. We could definitely provide empirical results demonstrating the effect of task ordering in the final version of the paper. Finally, we agree that task ordering and its impact on CLEMC is an interesting question and will be a good topic for future study.
>
>
>
> **Fig 1**  Figure 1 is a hierarchical visualization of the Dynamic Program based formulation of continual learning and the computation of the effective model capacity. At the lowest level we have the tasks. The model weights are updated based on the tasks using appropriate loss function minimization. The forgetting cost, $J_F$, for any task $k$ is obtained by minimizing the loss over tasks in the interval $[0, k]$. This is shown by the arrows in green. The value function, $V^*$, is the forgetting cost over all the tasks in $[1, K]$. This is shown by the arrows in black. The optimal capacity is computed as the sum of the effective capacities over tasks $[1, K]$ (shown in red).
> In response to this comment, we have updated the caption of Figure 1. We have also added text in Section 3, referencing Figure 1, as we discussed different parts of the figure.
>
> **Fig 6**  This was an interesting observation from our experiments. When we performed  experience replay (ER) for previous tasks along with a new task, we continued to observe an increase in forgetting cost in general, which supported our theory. Also, the cost with ER was slightly higher than without ER. We attributed this to the model learning a harder task due to the data mix. However, for the books task, when we performed an experience replay (and we repeated this experiment multiple times) we observed that the "with ER" cost was lower than the "without ER" cost. We believe that the model checkpoint from the previous task provided a better initialization for the books task, leading to lower cost. Also, it is possible that the books tasks had similarity with the previous  tasks dataset which helped in lowering the forgetting cost.
>
> [1] Doan, Thang, et al. "A theoretical analysis of catastrophic forgetting through the ntk overlap matrix." International Conference on Artificial Intelligence and Statistics. PMLR, 2021.

---

> > ### Comment · Reviewer_rQ9K · 2024-11-30
> >
> > I would like to thank the authors for addressing the questions and weaknesses raised, as well as for updating the manuscript. While I understand and appreciate the importance of establishing a theoretical framework, after reading the reviews from other reviewers, I would like to echo the concerns they have raised. In particular, I align with the concerns highlighted by reviewer hxVe regarding the key takeaways, motivations, and the potential for future work inspired by this framework. I would appreciate it if the authors could provide stronger justifications and clarifications on these points.

---

> ### Author Response · Authors · 2024-12-02
> **Response**
>
> Please see the collective response to all reviewers at the top of the page for clarification on these points.
> We hope this response has provided some clarity and you would reconsider your score.

---

> ### Author Response · Authors · 2024-12-03
>
> We thank you for the reviews and for your time. Since, the author response window closes tonight, we will not be able to answer any more questions.
>
> We hope that, our responses were satisfactory.

---

### Official Review · Reviewer_hxVe · 2024-11-09

**Soundness:** 3
**Presentation:** 2
**Contribution:** 2
**Rating:** 5
**Confidence:** 2

**Summary:**

This paper studies both theoretically and empirically the interaction between a model's capacity and continual learning. The paper first introduces CLEMC, a theoretical concept which describes the ability of a model to learn a new task continually without harming tasks learned in the past. The theory suggests that the forgetting cost grows with more continual tasks and capacity diverges (becomes worse) as more tasks are added. These findings are tested on MLP/CNN/GNN/LMMs.

**Strengths:**

S1: The topic of study is a very important one: Continual learning. The paper has a solid related works section motivating why this study is needed.

S2: The theory in the study is clear, and its conclusions are well highlighted.

S3: Experiments to confirm the predictions of the theory are done on different model architectures, which is essential to capture the generality of the theory.

**Weaknesses:**

W1: Some figures are unclear whether they support the claim. In specific, In Figure 4a, the capacity seems to drop back to a low value even though the initial bump gets bigger with more tasks. In Figure 5, there seems to be more changes seen in the weight, not explained by $\partial V^{*} / \partial x$.

W2: The FNN and GNN results could use more complex data to confirm its findings. While the authors mention that the experimental section is mostly meant for the ease of analysis, it is unclear how well these results will generalize to more complex settings. Can the authors show these results on realistic graph classification or MLP classification tasks?

W3: Although having a theoretical framework for CL is important, it isn't clear how surprising the finding from the theory are. It is unclear how surprising it is that 1) model hyperparameters (architecture, size, optimization, loss) affects the continual learning capacity and 2) more distant tasks causes a capacity degradation  and 3) the forgetting cost increases with more tasks. These finding seem intuitive. Are any of these claims surprising? If so, can this be discussed further?

W4: How does Figure 6 support its caption? Is wiki not treated as a task as it is close to the initial pre-training data? Should I interpret that the forgetting cost increase for the subsequent tasks?

W5: Figure 7,8 are hard to interpret. See Q1 below.

**Questions:**

Q1: It is quite hard to interpret Fig 7 and 8. Could a summary statistic be devised instead?

Q2: It would be nice to have highlighted claims for section 4.1: Theoretical analysis

Q3: This is very minor, but the wording "capacity" seems like a big values should be "good", i.e. more capacity, unlike what it is. Could this be clarified for quick skimming readers?

---

> ### Author Response · Authors · 2024-11-23
> **Response**
>
> We thank the reviewer for a positive review of the paper and for the valuable feedback and suggestions. Please find our responses below.
>
> **W1:** In Fig 4(a), we can see that the value of the CLEMC, $\epsilon^*_k$, grows as new tasks arrive -- indicating reduced capacity and supporting our theoretical claims.  In Figure 5, the changes in the task data occurs due to both node and edge feature changes in the graph. As new tasks arrive, we observe a change in weights  $\frac{\partial V^{*}}{  \partial w}$   to accommodate changes in either or both node features and edge features, that is, the partial derivative of the value function with respect to edge and node features.
>
> **W2:** We agree, but would like to humbly submit that in the CL community, Omniglot is a common dataset used for classification. The computational requirements for running the GNN on a large graph dataset makes it hard for us to run such experiments. Furthermore based on our experiments with relatively large language datasets we believe that the conclusions would most likely hold.
>
> **W3:** We completely agree with this observation. The claims by themselves are intuitive and unsurprising. Our goal was to create a theoretical framework (in the form of the Dynamic Program-based formulation of the CL problem) within which we could validate our intuition regarding these results and pin them to concrete terms. This connection would allow us to then further study the effect of these terms on capacity and forgetting and improve overall model performance.
>
> **W4:** In Figure 6, the pre-training cost curve is added as a reference. For pre-training a mix of all the data sets is used.Wiki is the first task on which the model is trained as mentioned in the setup. This is followed by git, arxiv and books. As we can see for both the 8M and the 134M parameter models, when the model is trained on wiki data alone, the forgetting cost is comparable to the pre-training cost (mostly because data is similar). As new tasks arrive, initially the cost continues to reduce, showing that the model is able to learn without forgetting. Eventually as more tasks arrive, with bigger differences in data we see that the forgetting cost increases. You can also observe that when data replay based regularization is performed the forgetting cost for validation improves.
>
> **Q1:** Figures 7 and 8 are meant to visually bring out the impact of new tasks impact the model weights (in terms of how much their magnitude changes). We have increased the size of the plots to improve readability. But, we can also add summary statistic about the distribution of weight changes in the final version of the paper.
>
> **Q2**  We have revised our text based on suggestions and comments received. The different headings in Section 4.1 have been modified to better highlight the claims and act as a summary of results.
>
> **Q3**  In this setup, big values are bad and small values are good. Based on these comments, we have now explicitly added a line in the introduction to make it clear that higher capacity is associated with larger forgetting cost and lower representational capability of the model.

---

> > ### Comment · Reviewer_hxVe · 2024-11-29
> >
> > >W1
> >
> > I understand your response for Fig 4. thank you.
> >
> >
> >  For Fig. 5, my question is asking the following:
> > The authors are trying to suggest that $\partial V^{*} / \partial x$ precedes the change in weights. However while this is clear from the big bump, where are unexplained dynamics in weights. How do these relate to dV/dx ? what weight changes are related and what are not?
> >
> > >W2
> >
> > Makes sense!
> >
> > >W3
> >
> > While I understand the value of such a theory, in the current form it isn't clear how this theory updates our view on CL and prompts specific experiments to pursue. My original review score was assuming that there *is* an explicit takeaway message from the theoretical framework or that the framework strongly suggests a specific experiment to pursue. I was assuming this was the case yet I didn't manage to grasp it (hence the confidence of 2). Unfortunately, I must update my score to 5 unless this issue is addressed.
> >
> > >W4
> >
> > I see, thank you. I now understand this experiment better. However, I think it shares the weakness W3.
> >
> >
> > ---
> > ---
> > >Q1
> >
> > Yes a summary statistic will help, especially since I don't see a need for a 2D image here. (No visual features)
> >
> > >Q2
> >
> > Thank you!
> >
> > >Q3
> >
> > Thank you!
> >
> > ---
> > ---
> >
> > **Summary:**
> >
> > The presentation of the manuscript has improved! However the major weaknesses, W1 (Fig 5) and most importantly W3 seems to remains. In my opinion, W3 really needs to be addressed in terms of:
> >
> > 1. What are the main takeaways from this framework?
> > 2. What experiments does this framework motivate?
> > 3. Why exactly does this framework allow an analysis which wasn't possible before?
> >
> > In the current form, I think I should update my score to 5.

---

> ### Author Response · Authors · 2024-12-02
>
> We have put a common response to all the reviewers on these as they concern the novelty and contributions of this paper.
>
> We hope this response have provided some clarity and you would reconsider your score.
>
> Thank you for your feedback and your time.

---

> ### Author Response · Authors · 2024-12-02
> **Fig. 5 (W1)**
>
> **For Fig. 5, my question is asking the following: The authors are trying to suggest that ∂V∗/∂x precedes the change in weights. However while this is clear from the big bump, where are unexplained dynamics in weights. How do these relate to dV/dx ? what weight changes are related and what are not?**
>  In practice, we would update the weights based on the data. Therefore, each jump in the edge or the node feature are followed by the jump in weights. This can be understood by the following three incremental steps \
> 	- Any change in task, brought about by a change in the edge $\phi$ and node $x$ features, reflects in a large jump in the forgetting cost $J_F$. \
> 	- A large jump in $J_F$ then leads to a change in the weights by virtue of the gradient necessary for minimizing $J_F,$ \
> 	- This change is then reflected in the weight updates.
>
> However, the dependence between weights and the tasks is recursive and mathematically inseparable (the first difference in capacity cannot be independently decomposed into the task and weight changes).  Mathematically, this can be seen by writing the first order KKT (Karush Kahn Tucker) conditions on $V^*$, which is equivalent to
> $$-\langle \partial_w V^*, dw \rangle = (V^*(k+1)-V^*(k)) + J_F +\langle \partial_x V^*, dx \rangle + \langle \partial_{\phi} V^*, d~\phi \rangle.$$
> where $k$ is the task index. it is clear from this equation that the dynamics between $x, \phi, w$ are not separable, i.e. $V^*$ implicitly depends on the weight, edge and node features. This recursive equation is how $x, \phi$ and $w$ relate to each other.

---

> > ### Author Response · Authors · 2024-12-03
> > **Thank you**
> >
> > We thank you for the reviews and for your time. Since, the author response window closes tonight, we will not be able to answer any more questions.
> >
> > We hope that, our responses were satisfactory.
> >
> > Regards
> > Authors

---

### Official Review · Reviewer_vBt4 · 2024-11-10

**Soundness:** 2
**Presentation:** 2
**Contribution:** 2
**Rating:** 5
**Confidence:** 3

**Summary:**

This paper provides a theoretical exploration of the relationship between model capacity and forgetting within a continual learning framework. The authors demonstrate that model capacity in CL is non-stationary, and the network's ability to represent or memorize new tasks diminishes as long as there is a distribution shift between tasks. This theoretical finding is supported by experiments conducted on various neural network architectures, including feed-forward, convolutional, graph neural networks, and transformer-based models.

**Strengths:**

1. The authors validate their theoretical results through extensive experiments using datasets relevant to the CL paradigm.
2. The derived theoretical results apply to various NN architectures, including feed-forward, convolutional, graph neural networks, and transformer-based models.

**Weaknesses:**

1. I went through the derivation of Lemma 1 and Theorem 1, which forms the basis for Theorems 2 and 3. However, I identified some potential issues that may have led to inaccuracies in the final results.

2. I believe the paper's presentation could be improved. There are several areas where the definitions regarding "capacity" appear to conflict, and important experimental details are missing, preventing the reader from fully understanding the results.

**Disclaimer**: It is possible that I may have misunderstood some of the derivations. If this is the case, I would appreciate a clear explanation from the authors. I am open to discussing and reconsidering my evaluation based on their responses. Please refer to the questions for more details.

**Questions:**

### Regarding derivation for Lemma 1 and Theorem 1
Please refer to the supplementary material for these questions unless stated otherwise.

First for Lemma 1:
1. **line 167, equation (6a), undefined term**. $\Delta V_{(j)}$ is not clearly defined. It seems to be higher-order terms, yet the authors defined $\Delta$ as the first-order difference so I am not sure. More importantly, moving from (6a) to (6b), it appears that  $\Delta V_{(j)}$ was assumed negligible by the authors, as it was omitted in (6b). If so, could the authors elaborate on what does $\Delta V_{(j)}$ represent and why is it negligible?
2. **line 167, equation (6a), Appropriateness of Taylor Expansion**. Is using a Taylor expansion at $V_{(j)}$ for approximating $V_{(j+1)}$ appropriate? Based on the definition of the value function $V$ e.g., Equation $(J_F)$ Equation (14), $V_{(j+1)}$ contains different terms in comparison to $V_{(j)}$, in the summations, therefore, they are not simply the same function at two different time steps?
3. **line 214, equation (9). Transition from (7) to (9)**. How was the transition made from (7) to (9) and ultimately to Lemma 1 (see line 196, main text). The minus sign on the left-hand side (LHS) of (7) appears to have been omitted when directly substituting (7) into (9) for the innermost minimization term. No adjustments seem to have been made to the two outer max/min operators or the order of k and k+1 on the LHS in Lemma 1 to account for this omission. Is the result correct? Additionally, was Lemma 3 a misreference, given that there is no Lemma 3 in the paper?

Assume Lemma 1 is still correct, for Theorem 1:

4. **line 252, Lemma. Definition of $\boldsymbol{k}$**. $\forall k\in \boldsymbol{k}$, was $\boldsymbol{k}$ defined somewhere before?
5. **line 263, deduction from Theorem 1**. The statement "then we know from Theorem 1 using the property $<a,b>=||a||||b||$" needs clarification. How was this deduced from Theorem 1 (and perhaps you mean Lemma 1)? The equality $<a,b>=||a||||b||cos(\theta)$, holds true when the two vectors are aligned. Can you elaborate on this deduction?
6. **line 269, missing closing and opening $||$ symbols for the two terms**?
7. **line 294, equation (15a) to (15b), Triangle Inequality**. Should the equality be replaced with a less than or equal to sign, as the norm of sum generally does not equal to the sum of norms based on Triangle inequality? And equation (15b) to (15c), inequality should be equality instead, as the norms without $w_{j}$ are all 0s?
8. **line 307,equation (16), Decay Factor**. The superscript for the decay factor $\gamma^{p}$ does not match the index used in summation (over $m$), is this correct?
9. **line 323, equation (17b), and line line 296, equation (15c), Implicit Dependency in optimization**. The authors derived these by stating that the norms in the summation without $w_{j}$ (the argument for which the derivate was taken) are all 0s. However, have we ignored the implicit dependency of $w_{j+1}$ on $w_{j}$ and likewise $w_{j+1}^*$ on $w_{j}^*$? The authors mentioned that $w_{j}^*$ is the starting point for optimizing $w_{j+1}^*$ when we consider the optimization trajectory in CL? If so, how is this justified?

These points need to be addressed to ensure the accuracy and clarity of the argument presented in Lemma 1 and Theorem 1, and consequently Theorems 2&3.

>

### Definition of "Effective Capacity" or "Capacity" in the Paper
The term "capacity" is used in multiple contexts, leading to potential confusion. It would be helpful if the authors could clarify the distinctions and connections between these definitions to ensure a consistent understanding throughout the paper.

- line 123, equation (EMC) first defines $\epsilon$ itself as the "effective model capacity"
- line 179: "as capacity at each $k$ is defined by the highest forgetting loss in the interval [0,k]". The description "as capacity at each k is defined by the highest forgetting loss in the interval [0,k]" suggests a different definition of capacity. It appears to refer to the term inside the summation in Equation CLEMC, introducing an extra outer maximization. This seems to deviate from the initial EMC definition in Definition 1.
- line 181:  The text mentions, " If the model learns ten tasks, then we obtain an EMC corresponding to each task, then, the $\epsilon_{(k)}^*$ is the sum of individual task capacities". This is confusing because it implies that individual task capacities (which are still seem to be defined using EMC) are summed to form the overall capacity, but this interpretation doesn't align clearly with the previous definitions.
- in line 243: The statement "Under these assumptions, we show that for both settings (i) and (ii) above, the effective capacity diverges". This "effective capacity" seem to refer to the gap in $\epsilon_{k}^*-\epsilon_{K}^*$ later as stated in line 250: "$\epsilon_{k}^*-\epsilon_{K}^*$ diverges"


>

### Empirical Observations and Their Alignment with Text and Theorems
It is possible that my current conclusions are a result of lacking detailed description on the experiments and setups. Clarification from the authors would be helpful to resolve these issues.

1. Figures. 2, 3, 4, how was $\epsilon_{k}^*$ calculated, was it strictly following Equation CLEMC? If so, based on Theorem 1 and 3, we expect an increasing (diverging) $\epsilon_{k}^*$, indicating that capacity gets poorer, for all $||\Delta T(p)||$, as long as $||\Delta T(p)||$ is non-zero. However, the green and blue lines in Figures 2B (top) and 3B (top) are decreasing over the training steps. This seems to contradict the expected theoretical trend.
2. Figure 2, what is $\Delta x$ exactly? I am aware that it is related to $||\Delta T(p)||$, as stated by the authors, but are these two quantities equivalent? For example, how is $\Delta x$ and $||\Delta T(p)||$ calculated in the sine experiments?
3. Line 333, "blue curve is better than the orange curve", is this referring to Fig 3, A? However, it's evident that the orange curves deteriorates slower then the blue ones, for all $||\Delta T(p)||$.
4. lines 346-353, analysis for Fig 4 A. I find the connection between the stability gap phenomenon (large jumps in losses at the start of each task) and Theorems 3 and 2 a bit unclear. Although the gaps increase over training steps, suggesting a loss of model capacity, the losses (or $\epsilon_{k}^*$  as plotted in A) quickly recovers, to a small value. Does this recovery conflict with the theoretical predictions?

>

### Minor comments on notations and typos
1. line 136-140, Equation $(J_F)$, main text. What is the feasible weight set $\mathcal{W}\_{(k)}$? Specifically, $T\_{(k)}=\\{T(0),T(1),....,T(k)\\}$ means the *collection* of all tasks until $k$, so I wonder whether $\mathcal{W}_{(k)}=\\{W(0),W(1),...,W(k)\\}$ is also a collection of all feasible sets until $k$ ? But based on Equation $(J_F)$ and the text that follow, it seems that $\mathcal{W}\_{(k)}$ represents only *one* specific feasible set at step $k$, with subscript $\_{(k)}$ for indexing. If my understand is correct, perhaps consider using $(k)$ and $k$ to distinguish between collection until $k$ and one specific index $k$ for improved clarity?
2. line 089, Equation $(CL)$, supplementary, an missing open bracket in the aurguments for the minimization problem. More importantly though, line 161, Equation $(CL)$, main text, should $w_{(i)}=\\{W_{(i)},i=k,k+1,...,K\\}$ be changed to $\\{w_{(i)}=W_{(i)},i=k,k+1,...,K\\}$ instead, since minimizing the sequence of $w_{(i)}$ is done with each $w_{(i)}$ being associated with its own feasible set i.e., $w_{(i)}\in W_{(i)}$?
3.  Why are the Lipschitz constants in square brackets, in Theorem 2 and 3? Are they interpreted as an argument of $alpha$, the learning rates, if written in this way?
4. Line 205, main text, maybe "effect" $\rightarrow$ "affect"?
5. Line 239 main text, "due to change in the data at the at the $p^{th}$ task", repeated "at the".
6. Supplementary Section B title: should refer to Lemma 1 instead of Theorem 1 and Supplementary Section C title should refer to Theorem 1 instead of Lemma 1.

---

> ### Author Response · Authors · 2024-11-23
> **Response  (Part 1)**
>
> We thank the reviewer for the extremely detailed and careful review of the manuscript. We realize that a lot of the confusion stemmed from the use of cumbersome notation and have tried to simplify it. Below we provide detailed responses to the various questions.
>
> # Lemma 1 and Theorem 1
>
> *line 167, Eq. 6(a):* Please observe that the derivation to Lemma 1 is motivated from the standard Hamilton Jacobi Bellman (HJB) equation's derivation and a nice intuitive version can be found in [1], Chapter 7. Indeed, in Eq. 6(a), the term $\Delta V_{(j)}$, does refer to all the higher order terms, i.e., terms corresponding to order second derivative and above. This is a standard practice in HJB derivation where the higher order terms are assumed negligible~[1] and the absence of these terms introduces an order of $2$ error into the approximation (Taylor approximation). Using $\Delta$ to denote the higher order terms was an error in notation. We have now replaced this notation with $\mathcal{O}(2)$ to indicate terms of order two and above (Supplementary Files, Eq. 1).
>
>
> *Appropriateness of Taylor Expansion:* The derivation in Lemma 1, follows the standard HJB derivation from the optimal control perspective, where it is assumed that all future information regarding the state space are known and kept constant. The only quantity that changes is the current cost. Under this construction, the additional term that comes in V(k+1) from k is $J(k).$ This is rather intuitive in the continuous time formulation as in [1].  To illustrate this,  without burdensome notation, we have the following alternative *informal* derivation of the HJB equation. Let
>
> $$V^*(k) = \underset{ \{ w(t), t = k, k+1, k+2, \dots \} }{min} \sum_{t=k}^{K} J(w(t))  $$
>
> where $w(t)$ represent the model weights and $J$ represent the forgetting cost. Then,
>
> $$V^*(k+1) = \underset{ \{ w(t), t = k+1, k+2, \cdots K \} }{min} \sum_{t=k+1}^{K} J(w(t))  $$
>
> By this structure, $V^*$ is a function of the weights starting at index $j$ and goes all the way to $K$. Just by subtracting, $V^*(k+1)-V^*(k),$ we can get
>
> $$ \Delta V^* = V^* (k+1)-V^* (k) = \underset{ \{ w(t), t = k+1, k+2, \cdots K \} }{min} \sum_{t=k+1}^{K} J(w(t))  - \underset{ \{ w(t), t = k, k+1, k+2, \dots \} }{min} \sum_{t=k+1}^{K} J(w(t)) $$
>
> which provides
>
> $$ \Delta V^* = V^* (k+1)-V^* (k) = \underset{ \{ w(t), t = k+1, k+2, \cdots K\} }{min} \left[ \cancel{\sum_{t=k+1}^{K} J(w(t))}   + J(w(k)) -  \cancel{ \sum_{t=k+1}^{K} J(w(t))} \right].$$
>
>
> This is under the assumption that all future information required to calculate $J$ at every future instance is known. Therefore, the exact difference between the optimal value function at $k$ and $k+1$ is the **forgetting cost $J(k).$** Ideally, making the optimal decision (weight value) at each $k$ should lead to an optimal sequence of weights for all task from $k+1$ to $K.$ However, in practice, exact cancellation will not happen and there will be a $\delta,$ a small perturbation around the current optimal cost. In particular,
> $$ \Delta V^*  = V^* (k+1)- V^* (k) = \underset{ \{ w(t), t = k+1, k+2, \cdots K\} }{min} \left[ \cancel{\sum_{t=k+1}^{K} J(w(t))}   + J(w(k)) -  \cancel{ \sum_{t=k+1}^{K} J(w(t))} + \delta \right] $$
> This $\delta$ are the perturbations to $V^* $ by addition of task at $k.$ These perturbations can be obtained by Taylor series of $V^* $ around $k$ given as $$ \langle \partial_x V^*, dx \rangle + \langle \partial_w V^*, dw \rangle + \mathcal{O}[2]$$ terms, ignoring the higher order terms provides the HJB. Please see if the revised derivation in the manuscript (**different from above**) introduces clarity (Supplementary Files, Section B, Lemma 1).
> *line 214, Eq. 9. Transition from Eq. 7 to Eq. 9:* Eq. 9 is actually not coming from Eq. 7 but rather a function of Eq. 8(b). When you multiply Eq. 9 with -1 and then, take the negative inside the first max and the second min, we have the substitution and eventually the final result. Multiplying by negative one does flip the mins and the max. We had omitted these steps in the current draft leading to this confusion. Reference to Lemma 3 is a typo.   We have re-written the proof in a step-by-step manner to make the above steps explicit and have removed any spurious references. (Supplementary Files, Eqs. 9 and 10).
> **line 252, Lemma. Definition of k.** k is used as an index for the incoming tasks throughout the text. $\mathbf{k} = [0, k]$ has been defined on line 84. However, we acknowledge that this could be done better. In the revised version, we state this definition upfront when we set up the notation. (Section 3, paragraph on notation setup).
> **line 263, <a,b>=||a||||b||cos(θ)** This was a typo, the inequality is $<a,b> \leq \|a\| \| b\|$. We have fixed it now. (Supplementary File, Section C, Cauchy-Schwarz inequality)

---

> ### Author Response · Authors · 2024-11-23
> **Response (Part2)**
>
> **line 294, Eq. 15(a) to Eq. 15(b):** You are absolutely correct, the equality in Eq.15(a) to Eq. 15(b) and inequality in Eq. 15(b) to Eq. 15(c) are switched. We have corrected this now. Note, that we use the Cauchy's inequality to make these deductions and these ideas should be clearer now. (Supplementary File, Section C, Eqs. 7(a) - 7(d)).
>
> **ine 307, Eq. 16:** That factor should be dependent on m and not p. We have corrected this. We also realized in the derivation that, making this contribution explicit, introduces confusion, therefore, we have revised the derivation to indicate this.
>
> **line 323, Eq. 17(b)** You are correct, there is an implicit dependency. However, we assume that all past information about this implicit dependency can be directly understood/observed by looking at $w^*_{j+1}.$ This is our primary assumption where a Markovian behavior is assumed in the weights and in the loss function. **Note that, without these assumptions, the derivations and theorems in this paper do not work. To ensure validity of this assumption, we write the forgetting cost in the inner sum as a repeating term for each new task. This constructions allows the validity of this assumption as seen in numerous reinforcement learning works.**
>
> ## Definition of "Effective Capacity" or "Capacity" in the Paper
>
> **line 123, (EMC):** This is to state the traditional manner in which prior works have defined effective model capacity by optimizing over possible data. The following paragraph also highlights that EMC is inadequate in the CL setting. Note here the $\epsilon$ does not have a task-specific subscript.
>
> **line 179: "as capacity at each k and line 181:** We have revised our text to separately define EMC, Forgetting EMC and Continual Learning EMC and clarify the intermediate steps.  At any given task, we perform minimization across all observed task~(the extra minimization), to get the forgetting cost. This is the summation over all previous tasks in $J_F$.  We use this to define the Forgetting EMC (FEMC) in **Definition 2**, which is the maximum forgetting cost over all the observed tasks. Finally, we define  CLEMC for a task k, in **Definition 3**, as the sum of FEMC over all tasks.
>
> **line 243:** The statement above refers to the term $\sum_{k}^{K} d \epsilon*_{k}$  where $d \epsilon_k*  $ is the first difference in CLEMC. These notions have been clarified now.
>
> ## Empirical Observations and Their Alignment with Text and Theorems
> **Figures. 2, 3, 4,..**  This behavior is actually expected because we are to an extent trying to reduce forgetting during training. However, the goal of these plots is to show that,  for any new task, the cost increases showing there is forgetting happening. As more tasks arrive we observe greater divergence in $\epsilon^*_k$. The model's learning is interrupted by the constant changes in the tasks.
> **Figure 2::** They are calculated as the Euclidean norm between the previous and the next task and these two notions are synonymous. In the revised text we have removed all references to $\Delta x$.
> **Line 333, "blue curve is** This recovery is expected because the model is learning and trying to reduce forgetting and in this process, you observe the stability gap. However, in the long run, the model is overwhelmed by the constant changes introduced by the tasks. This is the behavior being shown here.
> ## Minor comments on notations and typos
> **line 136-140** We have changed this notation. Now, both (k) and k refer to specific indexes, and $\mathbf{k} = [1, k]$ to the collection of observed tasks. Note, we now label tasks from $1, \cdots, K$ instead of $0,\cdots, K$.
> **line 089, Equation (CL)** We have modified these proofs and therefore the notations for clarity.
> **Why are the Lipschitz constants**This is basically the multiplication of the learning rate $\alpha$ with the Lipschitz constant. We have fixed this now with a parenthesis instead of square brackets. (Supplementary Files, Equations 24 and 25)
>
> **Line 239 main text,** Thank you for pointing this out. We have corrected it now.
>
> We hope that your questions were clarified. If you have any further questions, we are happy to explain more. Thank you again for your care and attention in reviewing this paper. Our paper has greatly benefitted from addressing the concerns raised and we genuinely appreciate your time.

---

> > ### Comment · Reviewer_vBt4 · 2024-12-01
> > **Thank you for your detailed response!**
> >
> > Dear Authors,
> >
> > I apologize for the late reply. I appreciate the time and effort the authors have dedicated to providing a detailed response to my review and making revisions to the original manuscript. My concerns regarding the derivations have been addressed and I have updated my rating for this.
> >
> > However, I still seek clarification on the connection between the stability gap and the experimental results supporting the theoretical findings presented in this paper. I understand that an increasing stability gap indicates a decreasing capacity of the model during continual learning. Additionally, I recognize that the loss is expected to decrease due to model training and optimization on each task in this context.
> >
> > My concern is whether this dynamic in optimization—where the stability gap occurs then followed by this decreasing loss—is fully captured by the theorem. It seems to me that the theorem primarily supports the overall increasing trend of loss, rather than detailed CL training dynamics. Is this interpretation correct?

---

> ### Author Response · Authors · 2024-12-02
> **Response**
>
> Thank you very much for your insights and sustained engagement with the paper. We have further added a clarification on the contributions and novelty of the paper through a common post to all authors.
>
>
> **My concern is whether this dynamic in optimization—where the stability gap occurs then followed by this decreasing loss—is fully captured by the theorem. It seems to me that the theorem primarily supports the overall increasing trend of loss, rather than detailed CL training dynamics. Is this interpretation correct?**
>
> The main conclusion of the theorem is essentially about the increasing trend of loss function, however, stability gap is the reason we see this increasing trend and this is observed in the proofs.
> We show in the proofs that, the stability gap introduced by a particular task will lead to divergence if the change in the task is too large and you can update only a finite number of times. This can be informally observed as follows. Note that the first difference in capacity is given (through Eq. FD in the paper) as
>
> $$ d \epsilon^*_{k}  = \langle \partial_T V^*, dT \rangle + \langle \partial_w V^*, dw \rangle.$$
>
> **We have ignored the max's and the min's and assumed number of tasks to be one to make notational simplifications (the generic form of this proof with less assumptions can be found in the paper, our main results).** Now, the change in capacity is a direct function of change introduced by $T, w$ which is the stability gap. That is, As soon as there is a change in T, there is a change in w and there is a change in capacity. If you add the left and right hand side for K steps (one task, $I$ updates), we get
>
> $$ \sum d \epsilon^*_{k}  = \sum \langle \partial_T V^*, dT \rangle + \sum \langle \partial_w V^*, dw \rangle.$$
>
> The left hand side is a telescopic sum across different updates on the same task and this reveals
>
> $$ \epsilon^*_{k+1} -  \epsilon^*_{k}  = \left[ \sum \langle \partial_T V^*, dT \rangle + \sum \langle \partial_w V^*, dw \rangle \right].$$
>
> Whether the stability gap introduced by a new task *(the first term on the right hand side)* is eliminated or not depends on whether the sum in the right hand side converges to zero or not. if it does not converge to zero, in otherwords, the change introduced by the tasks is so large that finite number of updates are insufficient for convergence, say $$ \left[ \sum \langle \partial_T V^*, dT \rangle + \sum \langle \partial_w V^*, dw \rangle \right] = d_k$$, then,
> $$ \epsilon^*_{k+1} -  \epsilon^*_{k}  = d_k.$$
>
> then, for a finite number of updates $T, $ this sum from 0 to K will diverge.
>
>
> We hope these responses have introduced clarity into the various aspects of the paper.

---

> ### Author Response · Authors · 2024-12-03
> **Thank you**
>
> We thank you for the reviews and for your time. Since, the author response window closes tonight, we will not be able to answer any more questions.
>
> We hope that, our responses were satisfactory.

---

### Author Response · Authors · 2024-11-23
**Summary Response**

Dear Reviewers,

Thank you  for recognizing the **novelty** and **contributions** in the paper. Moreover, thank you for pointing out that the paper could use some improved clarity in its exposition, notations and plots. We have therefore performed a major revision of the paper.

We have revised notations and simplified derivations for all the theoretical results. We have also attempted to improve the clarity of the writing and improved plots according to your suggestions.

Please use the uploaded revised version of the paper when reading the point to point responses below.

Thank you very much for your efforts and time, it genuinely improved the
paper.

*If the revision meets your satisfaction, please consider raising the scores, otherwise, please let us know if we can improve any specific parts with additional explanations.*

---

### Author Response · Authors · 2024-12-02
**Contributions and Novelty**

**Research Gap:** In CL, there is an implicit progression of forgetting with respect to tasks. Most of the literature attempt to reduct the impact of this progression using heuristics and empirical intuitions. While some of them are successful, they do not advance our understanding as to why forgetting evolves and how to counter this evolution. There are some theoretical work that do look at the change in the forgetting as a function of weight updates. However, they do so, mostly with stringent assumptions or in the two task case. Most importantly, capacity is often considered in literature as a static concept, which is not applicable to the CL because CL exhibits a dynamic evolution as a function of task, weights and the architecture.

**Why exactly does this framework allow an analysis which wasn't possible before?:**
This is the first time effective capacity in the CL setting has been defined. In fact, capacity has thus far been defined primarily as a function of the number of neurons or the number of data points without considering the effect of weights updates, which is of prime importance in CL because, each new task start with the weight parameters ideal for the previous tasks.

**Contribution**
 - **We are the first to define a notion of capacity in the CL sense (eq. CLEMC)**.
 - We then use the premise "CL is a problem that evolves dynamically with the tasks, thus capacity of CL needs to be studied within a dynamic framework." to be the **first to describe the evolution of capacity as difference equation (eq FD)**.
 - We further show **how distribution shift in tasks or the changes in the weights effect the capacity and how this effect evolves over the CL learning interval** (Theorem 1, 2, 3)
 - We finally verify all these theoretical insights through exhaustive experiments over diverse model architectures and data types.

**Future Experiments**
Given, that we describe the evolution of capacity concretely, there are several effects that can now studied precisely.
1. For instance, the effect of task ordering, variability in task distributions, the precise effect of hyper-parameters and model scale as well the effect of different types of weight update mechanism such as ADAM, ADAMW and so on (similar to the proof of Theorem 2).
2. From our theoretical framework we understand that, controlling the effect of different variables on capacity should improve the performance of a model in the CL setting. However, how precisely to measure the change in capacity, how to compensate for this change especially when the change in capacity evolves as a function of tasks has not been studied in this paper.
3. Moreover, our theoretical framework elucidates, how current CL approaches (ranging from regularization approaches to dynamically modifying the model architectures with more parameters) effect capacity.  Could we use the insights from the theoretical relationships to fundamentally improve CL performance (especially catastrophic forgetting) through judicious use/increase of the model capacity?

These are some of the perspectives that require large amount of experiments and further research to understand and develop new approaches to actually control the change in capacity. We argue that the perspectives need to be studied considering the dynamic evolution of the CL problem instead of the standard first order optimization perspective which is more common in the literature.

---

### Meta-Review · Area_Chair_RLNE · 2024-12-23

**Metareview:**

Despite various positive comments by reviewers and several score increases during the rebuttal, the average score of this submission is still below the acceptance threshold, with no reviewer strongly arguing in favour of the manuscript. As several important concerns still remain (in particular, clarity), this submission is unfortunately not yet ready for ICLR acceptance.

**Additional Comments On Reviewer Discussion:**

Overall a very positive Reviewer Discussion with several score increases and very detailed responses by the authors (including detailed derivations). No concerns here.

---

### Decision · Program_Chairs · 2025-01-22

Reject